# SALM4 negatively regulates NMDA receptor function and fear memory consolidation

Eunkyung Lie[1,2], Yeji Yeo[3], Eun-Jae Lee[4], Wangyong Shin [1], Kyungdeok Kim[1], Kyung Ah Han[5], Esther Yang[6], Tae-Yong Choi[7], Mihyun Bae[1], Suho Lee[1], Seung Min Um[3], Se-Young Choi [7], Hyun Kim[6], Jaewon Ko[5] & Eunjoon Kim [1,4 ✉]

Many synaptic adhesion molecules positively regulate synapse development and function, but relatively little is known about negative regulation. SALM4/Lrfn3 (synaptic adhesion-like molecule 4/leucine rich repeat and fibronectin type III domain containing 3) inhibits synapse development by suppressing other SALM family proteins, but whether SALM4 also inhibits synaptic function and specific behaviors remains unclear. Here we show that SALM4-knockout (*Lrfn3*−/−) male mice display enhanced contextual fear memory consolidation (7-day post-training) but not acquisition or 1-day retention, and exhibit normal cued fear, spatial, and object-recognition memory. The *Lrfn3*−/− hippocampus show increased currents of GluN2B-containing N-methyl-ᴅ-aspartate (NMDA) receptors (GluN2B-NMDARs), but not α-amino-3-hydroxy-5-methyl-4-isoxazole propionate (AMPA) receptors (AMPARs), which requires the presynaptic receptor tyrosine phosphatase PTPσ. Chronic treatment of *Lrfn3*−/− mice with fluoxetine, a selective serotonin reuptake inhibitor used to treat excessive fear memory that directly inhibits GluN2B-NMDARs, normalizes NMDAR function and contextual fear memory consolidation in *Lrfn3*−/− mice, although the GluN2B-specific NMDAR antagonist ifenprodil was not sufficient to reverse the enhanced fear memory consolidation. These results suggest that SALM4 suppresses excessive GluN2B-NMDAR (not AMPAR) function and fear memory consolidation (not acquisition).

[1] Center for Synaptic Brain Dysfunctions, Institute for Basic Science (IBS), Daejeon 34141, Korea. [2] Department of Chemistry, Dongguk University, Seoul 04620, Korea. [3] Department of Biological Sciences, Korea Advanced Institute for Science and Technology (KAIST), Daejeon 34141, Korea. [4] Department of Neurology, Asan Medical Center, University of Ulsan, College of Medicine, Seoul 05505, Korea. [5] Department of Brain and Cognitive Sciences, Daegu Gyeongbuk Institute of Science and Technology (DGIST), Hyeonpoong-Eup, Dalseong-Gun, Daegu 42988, Korea. [6] Department of Anatomy and Division of Brain Korea 21, Biomedical Science, College of Medicine, Korea University, Seoul 02841, Korea. [7] Department of Physiology and Neuroscience, Dental Research Institute, Seoul National University School of Dentistry, Seoul 03080, Korea. ✉email: kime@kaist.ac.kr

Synaptic adhesion molecules regulate various aspects of synapse development and function, influencing neural circuit assemblies and brain functions. A large number of synaptic adhesion molecules have been identified, with neuroligins and neurexins being prototypical molecules[1–15]. Many of these molecules positively regulate synapse development and function, but relatively little is known about negative regulation by these proteins, particularly of synapse functions such as postsynaptic receptor responses. One known example of negative regulation is that by MAM domain-containing glycosylphosphatidylinositol anchor (MDGA) family proteins, which interact with neuroligins and inhibit neuroligin-dependent synapse development[16–21].

Synaptic cell adhesion-like molecules (SALMs; also known as Lrfns) represent a family of leucine-rich repeat-containing synaptic cell adhesion molecules with five known members: SALM1/Lrfn2, SALM2/Lrfn1, SALM3/Lrfn4, SALM4/Lrfn3, and SALM5/Lrfn5[14,22–25]. Although all members share a similar domain structure, individual SALMs possess distinct functional features. Specifically, SALMs 1–3, but not SALM4 or SALM5, contain a C-terminal PDZ-binding motif that directly interacts with the PDZ domains of PSD-95, an excitatory scaffolding protein that is abundant in the postsynaptic density (PSD)[26,27]. SALM3 and SALM5, but not other SALMs, trans-synaptically interact with presynaptic LAR-family receptor tyrosine phosphatases (LAR-RPTPs) to promote excitatory and inhibitory synapse development[28–30], as recently detailed by X-ray crystallographic studies[31–34].

In addition to synapse development, SALMs regulate pre- and postsynaptic clustering of receptors and other adhesion molecules. SALM1 promotes dendritic clustering of N-methyl-D-aspartate (NMDA) receptors (NMDARs)[22], whereas SALM2 promotes synaptic localization of both NMDARs and α-amino-3-hydroxy-5-methyl-4-isoxazolepropionic acid (AMPA) receptors (AMPARs)[23]. SALM1 promotes presynaptic clustering of neurexin-1β in an F-actin- and phosphatidylinositol 4,5-bisphosphate-dependent manner[35], an action that may also affect postsynaptic receptor responses, given that neurexins trans-synaptically regulate NMDAR or AMPAR responses and NMDAR-dependent synaptic plasticity[36–38].

Intriguingly, postsynaptic SALM4 interacts in cis with other SALMs (SALM2, SALM3, and SALM5) and inhibits their functions[20]. Specifically, SALM4 suppresses SALM2-dependent facilitation of excitatory synapse development, as well as SALM3/5 trans-synaptic interactions with presynaptic LAR-RPTPs and SALM3/5-dependent presynaptic differentiation[20]. Intriguingly, presynaptic LAR-RPTPs have recently been shown to control presynaptic release[39] and postsynaptic NMDAR responses, but not AMPAR responses, through mechanisms not involving direct LAR-RPTP-dependent trans-synaptic adhesions[40,41]. However, whether SALM4 also negatively regulates synaptic functions such as postsynaptic NMDAR responses and any related behaviors remains unclear.

In the present study, we found that SALM4-knockout (KO) (Lrfn3−/−) mice displayed abnormally increased NMDAR-mediated, but not AMPAR-mediated, synaptic transmission involving the NMDAR GluN2B subunit and the presynaptic receptor tyrosine phosphatase PTPσ. This was associated with enhanced contextual fear memory consolidation (7-day post training), but not acquisition or 1-day retention, or changes in other forms of memory (cued fear, spatial, or object recognition). Treatment of Lrfn3−/− mice with fluoxetine, a medication used to treat excessive fear memory in humans such as posttraumatic stress disorders (PTSDs), which directly inhibit GluN2B-containing NMDARs (GluN2B-NMDARs), improved NMDAR-mediated currents and fear memory consolidation in Lrfn3−/− mice. These results suggest that SALM4 suppresses excessive NMDAR functions involving GluN2B and fear memory consolidation in mice.

## Results

### SALM4 deletion selectively enhances contextual fear memory consolidation.
To investigate in vivo functions of SALM4, we first examined behavioral abnormalities in Lrfn3−/− mice[20]. In a behavioral test designed to measure both contextual and cued fear memory, Lrfn3−/− mice showed normal fear memory acquisition on the first day (induced by both the spatial context and sound cue) and 24 h retention (induced by a sound cue in a different context [B not A]), but exhibited moderately increased 48 h fear memory retention (in the original spatial context [A]) (Fig. 1a, b). In another experiment designed to test only contextual (not cued) fear memory, Lrfn3−/− mice displayed normal fear memory acquisition and 24 h fear memory retention but moderately enhanced 7-day fear memory consolidation (Fig. 1c–f). Here, fear memory lasting for 7 days was defined as consolidated memory, based on previous results[42,43], although we did not directly test it in the present study by using, i.e., protein synthesis inhibitors. In addition, Lrfn3−/− mice showed normal extinction of acquired contextual fear memory, although there is a decreasing tendency (Fig. 1g, h). These results suggest that SALM4 deletion enhances contextual fear memory consolidation without affecting acquisition or 1-day retention or cued fear memory.

In the Morris water maze test, Lrfn3−/− mice performed normally in the acquisition, probe, and reversal phases of the test (Fig. 1i), although the levels of performance were low for unclear reasons, and we did not attempt other versions of the Morris water maze test[44]. Lrfn3−/− mice also performed normally in the novel object-recognition test (Fig. 1j). In addition, Lrfn3−/− mice showed no changes in tests measuring locomotion, motor learning, social interaction, sensory-motor coordination, repetitive behaviors, anxiety-like behavior, or depression-like behavior (Supplementary Figs. 1 and 2). These results collectively suggest that SALM4 deletion selectively enhances contextual fear memory consolidation but not acquisition or 24 h retention.

### Increased NMDAR, but not AMPAR, currents, and normal synaptic plasticity at hippocampal synapses of juvenile and adult Lrfn3−/− mice.
To test whether altered synaptic transmission might underlie the enhanced fear memory consolidation in Lrfn3−/− mice, we measured AMPAR- and NMDAR-mediated synaptic transmission in the hippocampus, a brain region that, together with the amygdala, contributes to contextual fear memory[45–47]. We found that NMDAR-mediated (but not AMPAR-mediated) evoked synaptic transmission was increased (~53%) at Schaffer collateral-CA1 pyramidal (SC-CA1) synapses in juvenile Lrfn3−/− mice (postnatal [P] days 21–24), as shown by the ratio of NMDAR to AMPAR excitatory postsynaptic currents (EPSCs) (Fig. 2a). This, in combination with normal AMPAR-EPSCs at juvenile Lrfn3−/− SC-CA1 synapses[20], indicates an increased NMDAR component of excitatory synaptic transmission. In addition, the increase in NMDAR-EPSCs was mainly attributable to an increase in currents mediated by GluN2B-containing NMDARs (~50%), as shown by the increased sensitivity of the mutant currents to the GluN2B-specific inhibitor, ifenprodil (ifenprodil-sensitive NMDAR currents being 33% vs. 53% of total currents in wild-type (WT) and mutant neurons, respectively) (Fig. 2b), which indicates a 150% increase in GluN2B-NMDAR currents but a 25% decrease in GluN2A-NMDAR currents.

As shown by field recordings, this increased NMDAR-mediated synaptic transmission persisted into adulthood in Lrfn3−/− mice (P68–79; Fig. 2c) and involved increased postsynaptic responses

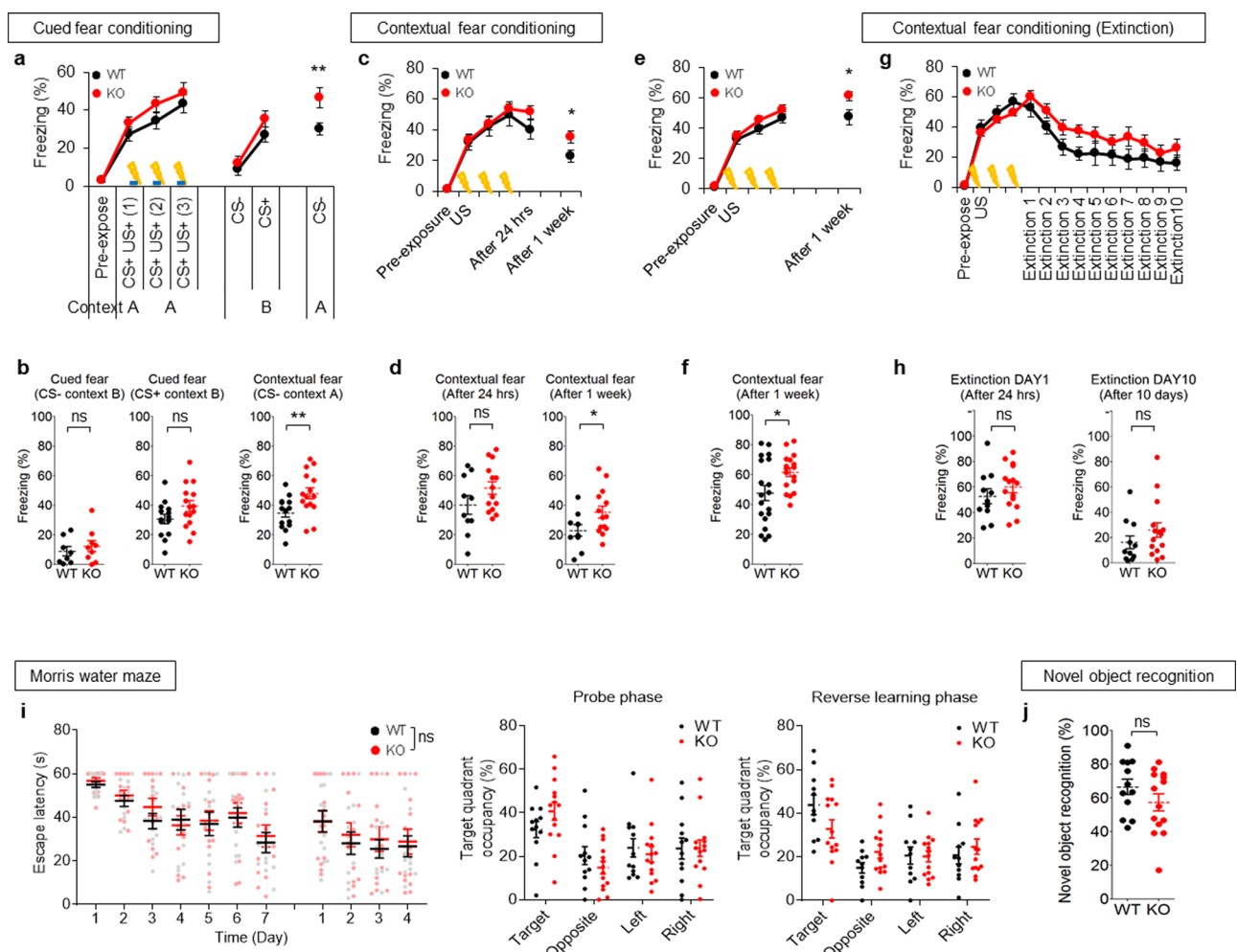

**Fig. 1 $Lrfn3^{-/-}$ mice show enhanced contextual fear memory consolidation but normal fear memory acquisition and 1-day retention, cued fear memory, spatial learning and memory, and object-recognition memory. a** Normal contextual fear memory acquisition and normal cued fear memory retention (1-day post training), but enhanced contextual fear memory retention (2-day post training), in $Lrfn3^{-/-}$ mice (8–12 weeks). Mice were introduced into a fear-conditioning chamber and exposed to a cue sound on day 1 for fear conditioning (by both context [context A] and sound cue), then exposed to a different chamber (context B) with the same sound on day 2 for cued fear memory (1-day retention), followed by re-exposure to the original chamber [context A] without sound on day 3 for contextual fear memory (or 2-day retention or memory consolidation). ($n = 14$ mice [wild-type (WT)] and 15 [knockout (KO)], two-way repeated-measure (RM-ANOVA) [no genotype difference], **$p < 0.01$ indicate the results of Student's test in **b**). **b** Quantification of the results in **a**. ($n = 14$ mice [wild-type (WT)] and 15 [knockout (KO)], **$p < 0.01$, ns, not significant, Student's test). **c** Enhanced contextual fear memory consolidation on day 8, but not on day 1, in $Lrfn3^{-/-}$ mice (20–24 weeks). Mice fear conditioned on day 1 were re-exposed successively to the same chamber on day 2 and day 9 for 24 h fear memory retention and 7-day fear memory consolidation, respectively. ($n = 10$ [WT] and 14 [KO], two-way RM-ANOVA [no genotype difference], *$p < 0.05$ indicate the results of Student's test in **d**). **d** Quantification of the results in **c**. ($n = 10$ [WT] and 14 [KO], *$p < 0.05$, ns, not significant, Student's test). **e** Enhanced contextual fear memory consolidation on day 8 in $Lrfn3^{-/-}$ mice (8–12 weeks). Mice fear conditioned on day 1 were re-exposed to the same chamber on day 8. ($n = 19$ [WT] and 17 [KO], two-way RM-ANOVA [no genotype difference], *$p < 0.05$ in the graph indicates the results of Student's test in **f**). **f** Quantification of the results in **e**. ($n = 19$ [WT] and 17 [KO], *$p < 0.05$, Student's test). **g** Normal contextual fear extinction in $Lrfn3^{-/-}$ mice (8–12 weeks). Fear extinction was tested by re-exposing mice fear conditioned on day 1 to the same chamber for 10 days starting from day 2. ($n = 11$ [WT] and 15 [KO], two-way RM-ANOVA [no genotype difference]). **h** Quantification of the results in **g**. ($n = 11$ [WT] and 15 [KO]; ns, not significant, Student's t-test). **i** Normal performance of $Lrfn3^{-/-}$ mice (14–22 weeks) in the learning, probe, and reversal phases of the Morris water maze test. ($n = 12$ [WT] and 14 [KO]; two-way RM-ANOVA (no genotype difference for both learning and reversal phases); ns, not significant, Student's t-test (quadrant occupancy in the initial and reversal probe phases)). **j** Normal performance of $Lrfn3^{-/-}$ mice (10–14 weeks) in the novel object-recognition test. One of the two identical objects presented to the subject mouse was replaced with a new object on the next day. ($n = 12$ [WT] and 14 [KO]; ns, not significant, Student's t-test). Error bars represent the SEM.

rather than presynaptic axonal conductance or nerve-terminal fiber volley responses (Fig. 2d, e). This change was not associated with altered paired-pulse facilitation or input–output curve (Supplementary Fig. 3a, b), suggestive of normal presynaptic release. There were also no changes in synaptic plasticity measures, including theta-burst stimulation-induced long-term potentiation (TBS-LTP), high-frequency stimulation (100 Hz)-induced LTP

(HFS-LTP), and low-frequency stimulation-induced long-term depression (LFS-LTD; single-pulse LFS (SP-LFS) at 1 Hz for 15 min) at SC-CA1 synapses of $Lrfn3^{-/-}$ mice (P56–63) (Supplementary Fig. 3c–h). In addition, there was no genotype difference in the HFS-LTP at temporoammonic (TA; perforant path)-CA1 synapses in the distal region of the CA1 subfield (Supplementary Fig. 3i–l). Synaptic plasticity at other hippocampal

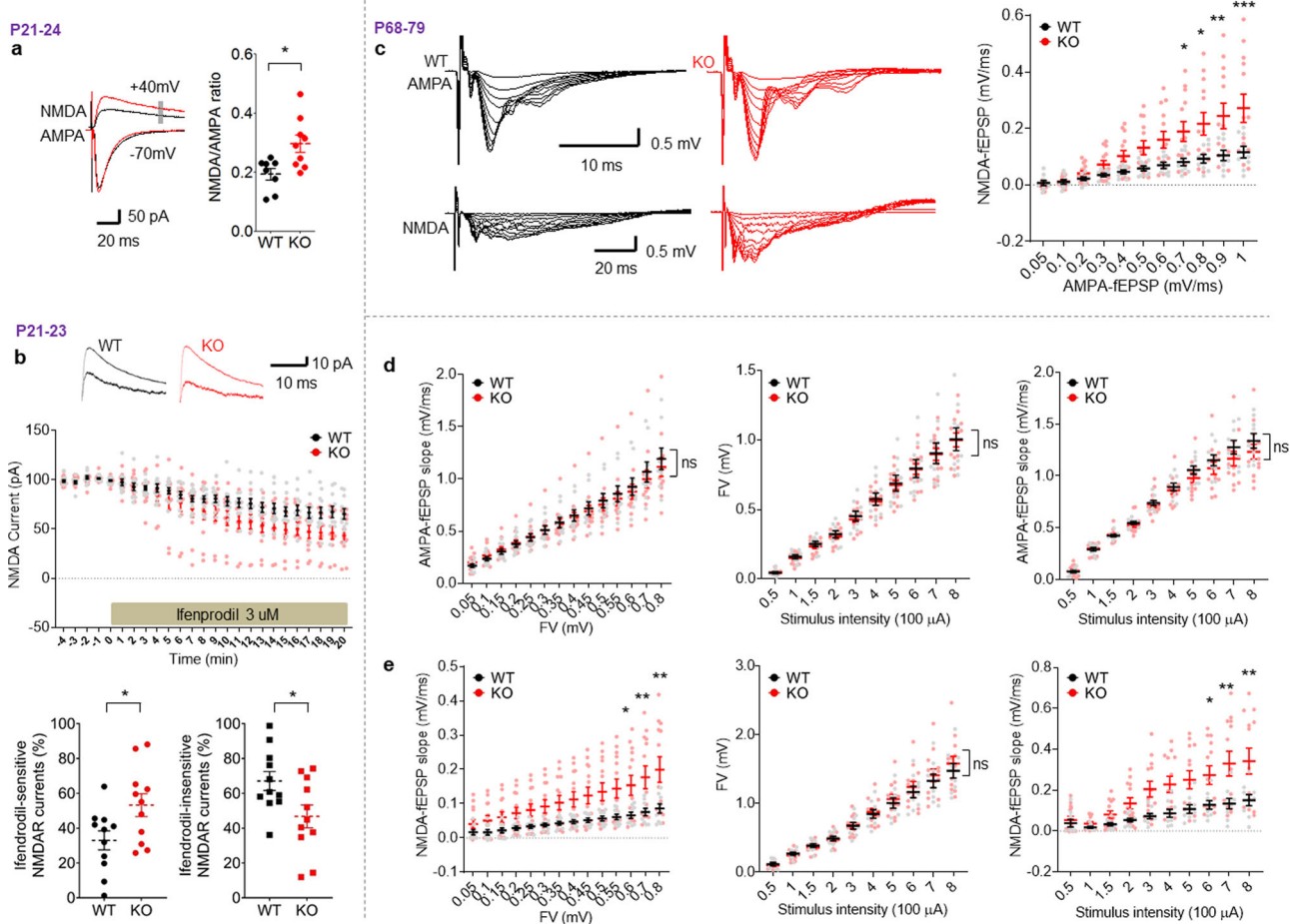

**Fig. 2 Increased NMDAR, but not AMPAR, currents at hippocampal synapses of juvenile and adult *Lrfn3*⁻/⁻ mice. a** Increased NMDAR currents at SC-CA1 synapses in the hippocampus of juvenile *Lrfn3*⁻/⁻ mice (P21–24), as indicated by the ratio of NMDAR- to AMPAR-EPSCs. ($n = 8$ slices from 6 mice [WT] and 9, 5 [KO]; *$p < 0.05$, Student's *t*-test). **b** Increased currents of GuN2B-containing NMDARs at SC-CA1 synapses of *Lrfn3*⁻/⁻ mice (P21–23), as shown by the sensitivity of the currents to ifenprodil (GluN2B-specific antagonist). ($n = 11$ neurons from 6 mice [WT] and 11, 5 [KO]; *$p < 0.05$ (last five points), Student's *t*-test). **c** Increased ratio of NMDAR- to AMPAR-mediated synaptic transmission at SC-CA1 synapses in adult *Lrfn3*⁻/⁻ mice (P68–79), as indicated by the initial slopes of NMDAR-fEPSPs plotted against those for AMPAR-fEPSPs. ($n = 9$ slices from 5 mice [WT] and 13, 5 (KO); *$p < 0.05$, **$p < 0.01$, ***$p < 0.001$, two-way RM-ANOVA with Bonferroni test). **d** Dissection of AMPAR-fEPSPs by plotting initial fEPSP slopes against fiber volley amplitudes (left) or stimulation intensities (right), or plotting fiber volley amplitudes against stimulation intensities (middle). ($n = 9$, 5 [WT] and 13, 5 [KO]; ns, not significant, two-way RM-ANOVA). **e** Dissection of NMDAR-fEPSPs by plotting initial fEPSP slopes against fiber volley amplitudes (left) or stimulation intensities (right), or plotting fiber volley amplitudes against stimulation intensities (middle). ($n = 9$, 5 [WT] and 13, 5 [KO], two-way RM-ANOVA [genotype p for left/middle/right panels are 0.03/0.71/0.02]; *$p < 0.05$ and **$p < 0.01$ indicate results of Bonferroni tests). Error bars represent the SEM.

subfields such as dentate gyrus and CA3 regions was not measured, although we could detect SALM4/*Lrfn3* mRNAs in other hippocampal regions, including the dentate gyrus and CA3 (Supplementary Fig. 4). These results collectively suggest that a SALM4 deficiency leads to selective and persistent increases in NMDAR-mediated, but not AMPAR-mediated, synaptic transmission at hippocampal synapses in juvenile and adult mice, without affecting synaptic plasticity, likely through compensatory changes (see below).

**NMDAR modulators differentially modulate NMDAR function and fear memory in WT and *Lrfn3*⁻/⁻ mice.** We next tested whether the increase in NMDAR currents could be responsible for the enhanced fear memory consolidation in *Lrfn3*⁻/⁻ mice by pharmacologically modulating NMDARs. To this end, we first treated naive WT and *Lrfn3*⁻/⁻ mice chronically with the NMDAR agonist D-cycloserine (DCS) (20 mg/kg/d; 7 days; intraperitoneally (i.p.)) or the NMDAR antagonist

memantine (10 mg/kg/d; 7 days; i.p.) and measured NMDAR-mediated synaptic transmission at SC-CA1 synapses in hippocampal slices. We used a chronic drug treatment scheme, because these drugs have short half-lives, and our goal was to obtain maximal drug effects. In addition, we used both DCS and memantine, because there was the possibility that chronic drug treatments could lead to opposite responses through receptor desensitization or compensatory responses.

At WT synapses, chronic DCS and memantine treatments induced trends (an increase and a decrease, respectively) in NMDAR currents, although only the DCS-dependent increase was statistically significant (Fig. 3a–g and Supplementary Fig. 5). At *Lrfn3*⁻/⁻ synapses, memantine induced a similar trend toward decreased NMDAR currents, whereas DCS unexpectedly tended to decrease NMDAR function, although this effect was not significant. The distinct effects of DCS and memantine might involve differences in dosage/treatment scheme, action mechanisms (glycine-site antagonist and open-channel blocker, respectively), or sensitivity of distinct neuronal types and synapses[48–50].

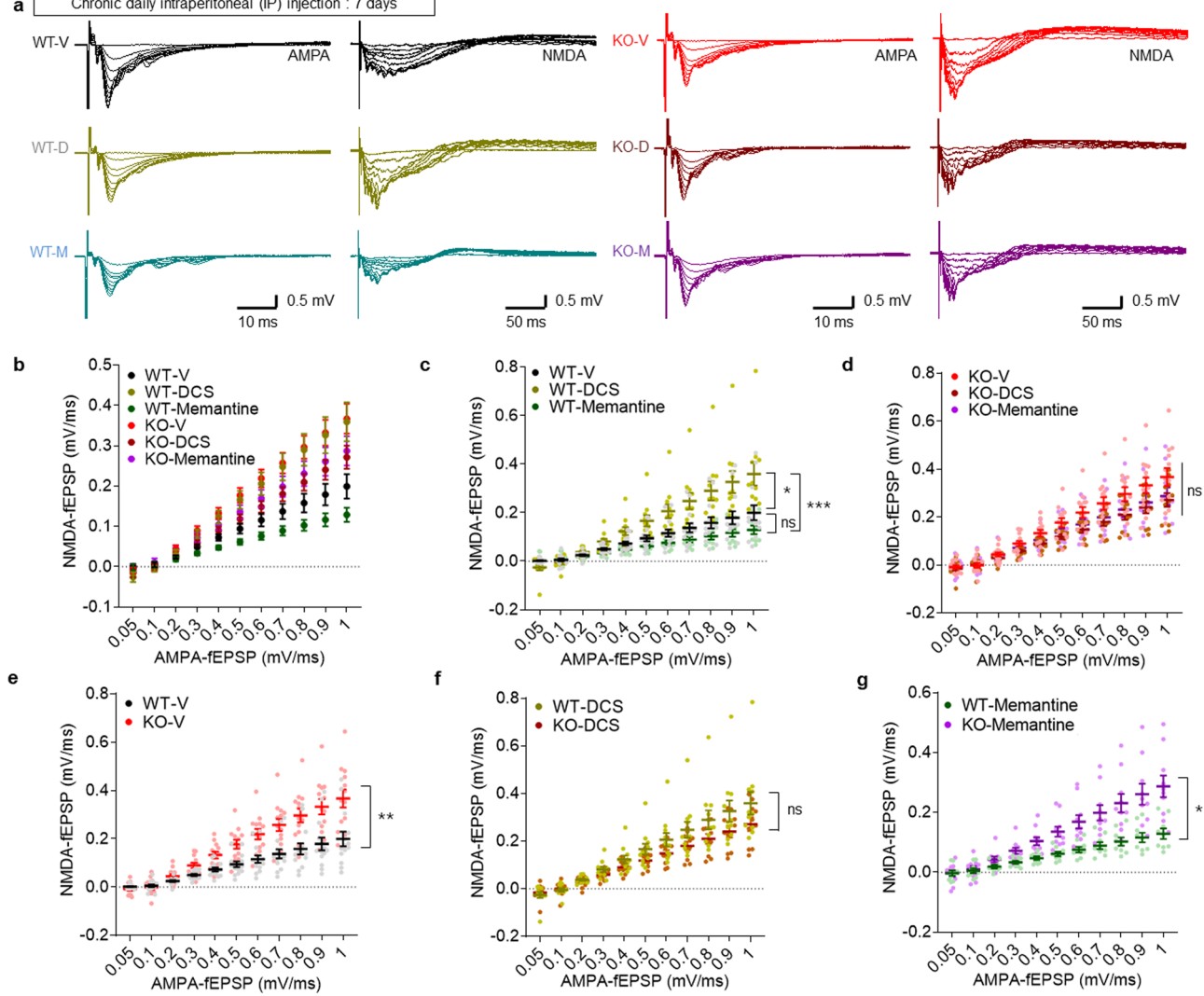

**Fig. 3 Differential effects of NMDAR modulators on NMDAR and AMPAR functions at *Lrfn3*$^{-/-}$ hippocampal synapses. a** Sample traces for the changes in NMDAR and AMPAR components of excitatory synaptic transmission at SC-CA1 synapses induced by chronic treatment of adult *Lrfn3*$^{-/-}$ mice (P56–91) with DCS (20 mg/kg/d; 7 days; i.p.), memantine (10 mg/kg/d; 7 days; i.p.), or vehicle (V), as shown by NMDAR-fEPSP slopes plotted against AMPAR-fEPSP slopes. **b** Quantification of the results in **a**. ($n = 14$ slices from 4 mice [WT-V], 13, 4 [KO-V], 11, 3 [WT-DCS], 10, 3 [KO-DCS], 9, 3 [WT-Memantine], and 10, 3 [KO-Memantine]). **c** A subset of the results displayed in **b** showing the effects of DCS and memantine on WT mice. ($n = 14$, 4 [WT-V], 11, 3 [WT-DCS], and 9, 3 [WT-Memantine], *$p < 0.05$, ***$p < 0.001$, ns, not significant, two-way RM-ANOVA with Bonferroni test [the statistical results are also a subset of the total two-way ANOVA results]). **d** A subset of the results displayed in **b** showing the effects of DCS and memantine on KO mice. ($n = 13$, 4 [KO-V], 10, 3 [KO-DCS], and 10, 3 [KO-Memantine]; ns, not significant, two-way RM-ANOVA with Bonferroni test). **e** A subset of the results displayed in **b** comparing vehicle-treated WT and KO mice. ($n = 14$, 4 [WT-V] and 13, 4 [KO-V], **$p < 0.01$, two-way RM-ANOVA with Bonferroni test). **f** A subset of the results displayed in **b** comparing DCS-treated WT and KO mice. ($n = 11$, 3 [WT-DCS] and 10, 3 [KO-DCS]; ns, not significant, two-way RM-ANOVA with Bonferroni test)). **g** A subset of the results displayed in **b** comparing memantine-treated WT and KO mice. ($n = 9$, 3 [WT-Memantine] and 10, 3 [KO-Memantine], *$p < 0.05$, two-way RM-ANOVA with Bonferroni test). Error bars represent the SEM.

We also tested whether these chronic drug treatments were capable of improving enhanced fear memory consolidation in *Lrfn3*$^{-/-}$ mice, which showed normal acquisition of contextual fear memory (Fig. 4a, b). Chronic memantine treatment failed to rescue the enhanced fear memory in *Lrfn3*$^{-/-}$ mice (Fig. 4c–g). Similarly, chronic ifenprodil, an antagonist for GluN2B-containing NMDARs, failed to rescue the enhanced fear memory consolidation in *Lrfn3*$^{-/-}$ mice (Supplementary Fig. 6). Intriguingly, DCS tended to decrease the enhanced fear memory in *Lrfn3*$^{-/-}$ mice, although this effect did not reach statistical significance. The strong DCS-dependent tendency toward normalization of the NMDAR and fear memory phenotypes, although inconclusive, suggests that NMDAR

modulation could be a useful strategy to correct abnormal phenotypes in *Lrfn3*$^{-/-}$ mice.

**Fluoxetine normalizes NMDAR function and fear memory in *Lrfn3*$^{-/-}$ mice.** Selective serotonin reuptake inhibitors, such as fluoxetine, are used to treat excessive fear memory in humans with PTSD[51–53]. In addition, fluoxetine directly inhibits GluN2B-containing, but not GluN2A-containing, NMDARs[54,55], although it also acts as a selective serotonin reuptake inhibitor. We thus tested whether fluoxetine treatment was capable of normalizing NMDAR hyperfunction and enhanced fear memory consolidation in *Lrfn3*$^{-/-}$ mice.

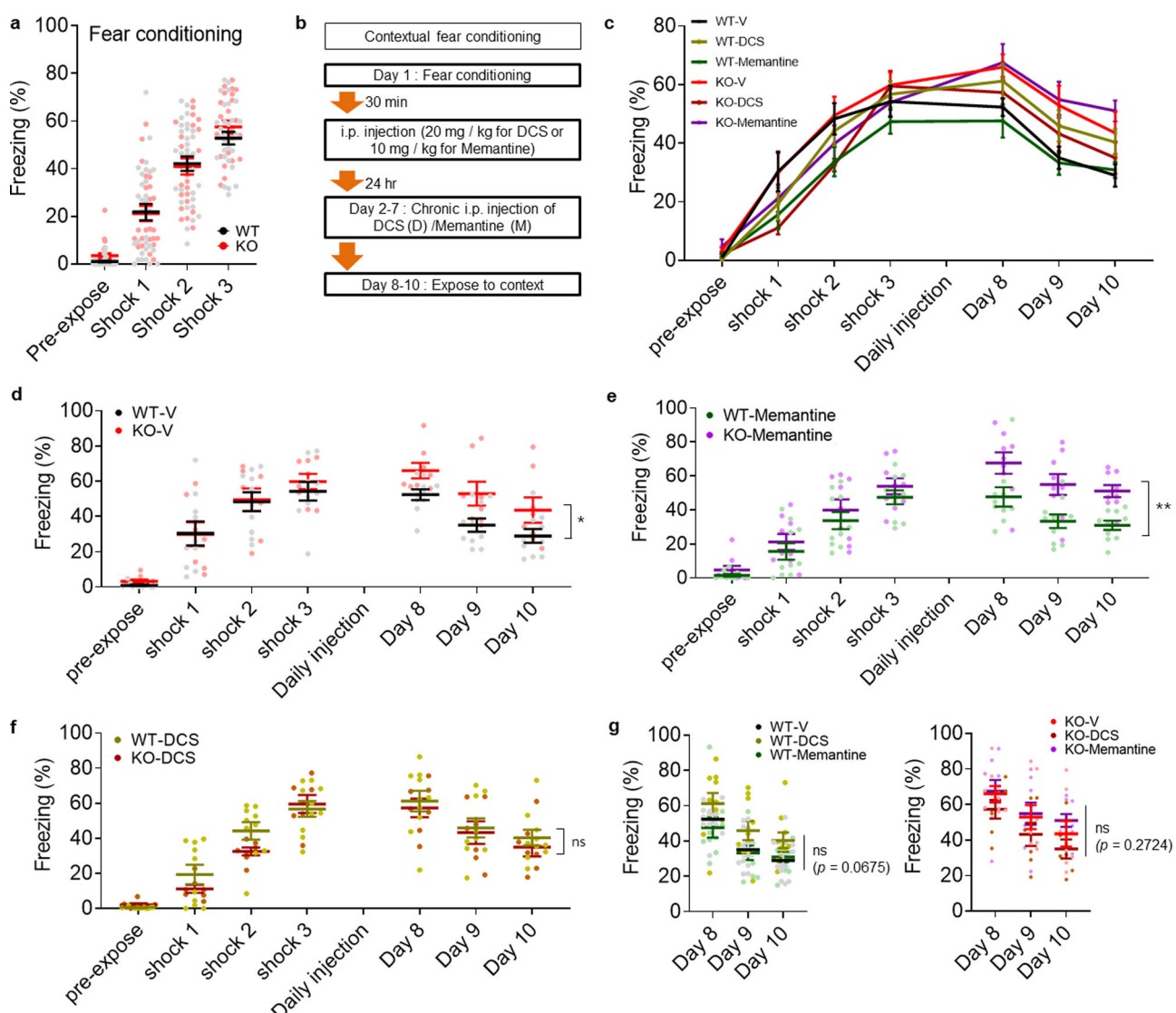

**Fig. 4 Differential effects of NMDAR modulators on fear memory consolidation in *Lrfn3*$^{-/-}$ mice. a** Normal acquisition of fear memory in adult *Lrfn3*$^{-/-}$ mice (P56–91). ($n = 30$ mice [WT] and 24 [KO], two-way RM-ANOVA [genotype $p = 0.6441$]). **b** Experimental scheme for chronic treatments of DCS (D; 20 mg/kg/d; 7 days; i.p.), memantine (D; 20 mg/kg/d; 7 days; i.p.), or vehicle (V) after fear acquisition and before the test of contextual fear memory consolidation, as measured by freezing levels during days 8–10. After initial contextual fear memory acquisition (**a**), WT and KO mice were divided into six groups (WT-V/D/M and KO-V/D/M) for chronic drug treatments. **c** Summary of the quantification of the results from **a** and **b**. ($n = 10$ mice [WT-V], 10 [WT-D], 10 [WT-M], 8 [KO-V], 7 [KO-D], and 9 [KO-M]). **d** A subset of the results displayed in **c** comparing WT-V and KO-V. ($n = 10$ [WT-V], 8 [KO-V], *$p < 0.05$, two-way RM-ANOVA [the statistical results are also a subset of the total two-way ANOVA results]). **e** A subset of the results displayed in **c** comparing WT-M and KO-M. ($n = 10$ [WT-V], 9 [KO-V], **$p < 0.01$, two-way RM-ANOVA). **f** A subset of the results displayed in **c** comparing WT-D and KO-D. ($n = 10$ [WT-V], 7 [KO-V]; ns, not significant, two-way RM-ANOVA). **g** Subsets of the results displayed in **c** comparing WT-V/D/M or KO-V/D/M. ($n = 10$ mice [WT-V], 10 [WT-D], 10 [WT-M], 8 [KO-V], 7 [KO-D], and 9 [KO-M]; ns, not significant, two-way RM-ANOVA). Error bars represent the SEM.

Chronic treatment of *Lrfn3*$^{-/-}$ mice with fluoxetine (5 mg/kg/d; 7 days; i.p.) significantly decreased NMDAR-mediated, but not AMPAR-mediated, synaptic transmission at *Lrfn3*$^{-/-}$ hippocampal SC-CA1 synapses, as shown by the significant difference in NMDAR-fEPSP/AMPAR-fEPSP ratios and field excitatory postsynaptic potential (fEPSP) slopes plotted against stimulus intensities between KO-vehicle and KO-fluoxetine groups (Fig. 5a–c). WT synapses also showed an NMDAR-specific fluoxetine-dependent decrease, in line with previous results[54,55].

Importantly, fluoxetine treatment normalized enhanced fear memory consolidation in *Lrfn3*$^{-/-}$ mice, as supported by the significant difference between KO-fluoxetine and KO-vehicle

groups (day 8) (Fig. 6a–d). Fluoxetine-treated WT mice showed a similar decrease in fear memory consolidation (day 8), in line with a previous report[56]. The effects of fluoxetine on days 9 and 10 were not significant in WT or *Lrfn3*$^{-/-}$ mice, with the freezing levels being similar across days 8–10, suggesting that fluoxetine has bigger effects on fear recall. The lack of baseline difference in freezing levels between vehicle-treated WT and vehicle-treated *Lrfn3*$^{-/-}$ mice, which differ from the results in Fig. 1e, f, could be attributable to that the experiment for Fig. 1e, f used naive mice, whereas that for Fig. 6a–d used mice handled and drug-treated for 7 days. These results collectively suggest that fluoxetine normalizes both NMDAR hyperfunction and enhanced fear memory consolidation in *Lrfn3*$^{-/-}$ mice.

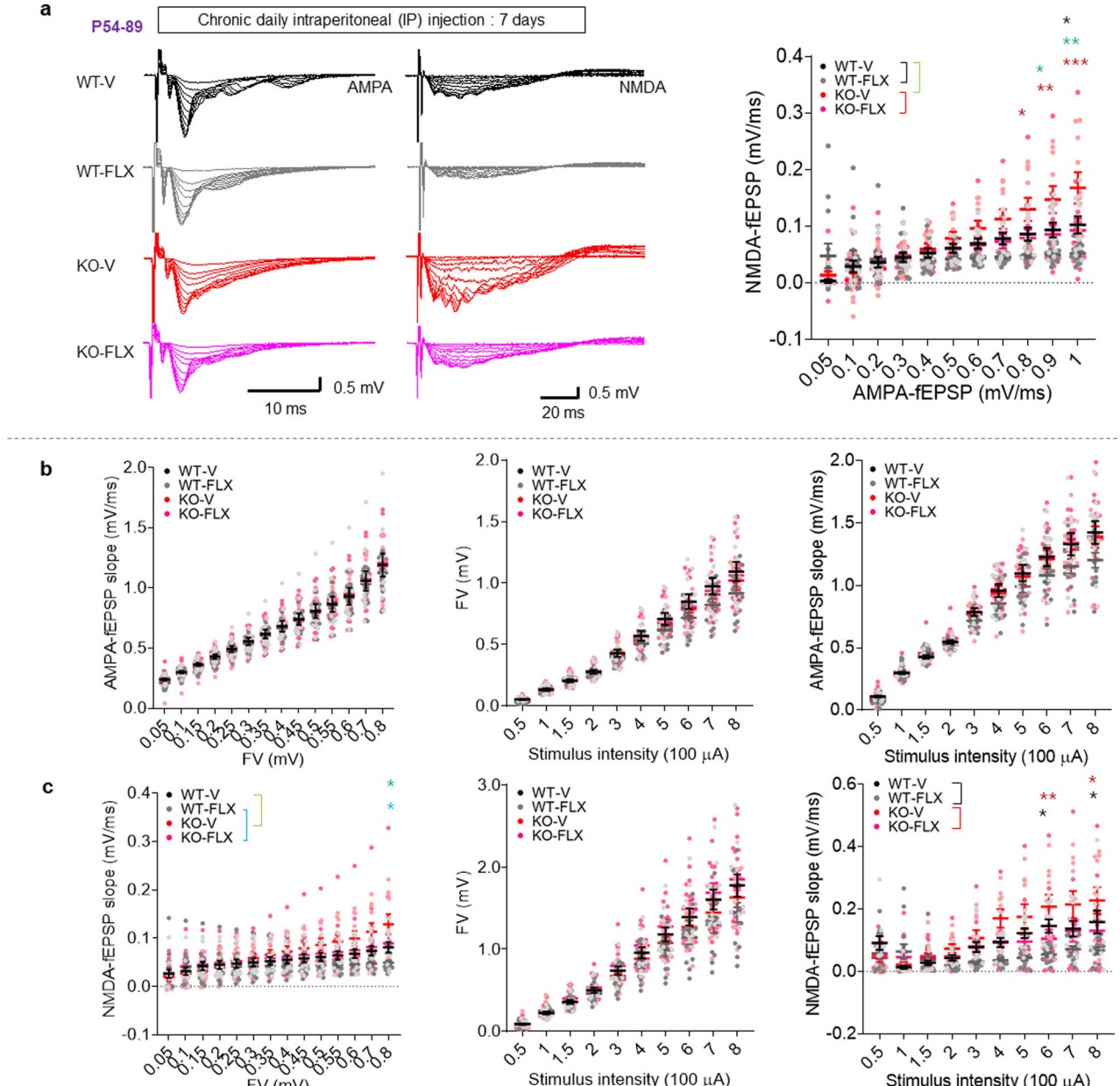

**Fig. 5 Fluoxetine normalizes NMDAR function in *Lrfn3*$^{-/-}$ mice. a** Chronic fluoxetine treatment (5 mg/kg/d; 7 days; i.p.) reduces the ratio of NMDAR- to and AMPAR-mediated synaptic transmission at SC-CA1 synapses in the hippocampus of *Lrfn3*$^{-/-}$ mice (P54-89), as indicated by NMDAR-fEPSP slopes plotted against AMPAR-fEPSP slopes. ($n = 11$ slices from 5 mice (WT-V/vehicle), $n = 13$, 6 (WT-FLX/fluoxetine), $n = 10$, 5 (KO-V), and $n = 15$, 5 (KO-FLX); *$p < 0.05$, **$p < 0.01$, ***$p < 0.001$, two-way RM-ANOVA with Bonferroni test). **b** Chronic fluoxetine treatment does not affect AMPAR-fEPSPs at *Lrfn3*$^{-/-}$ SC-CA1 synapses, as shown by plotting initial AMPAR-fEPSP slopes against fiber volley amplitudes (left) or stimulation intensities (right), or plotting fiber volley amplitudes against stimulation intensities (middle). ($n = 11$, 5 [WT-V], $n = 13$, 6 [WT-FLX], $n = 10$, 5 [KO-V], and $n = 15$, 5 [KO-FLX], two-way RM-ANOVA [no genotype difference]). **c** Chronic fluoxetine treatment reduces NMDAR-fEPSPs at *Lrfn3*$^{-/-}$ SC-CA1 synapses, as shown by plotting initial NMDAR-fEPSP slopes against fiber volley amplitudes (left) or stimulation intensities (right), or plotting fiber volley amplitudes against stimulation intensities (middle). ($n = 11$, 5 [WT-V], $n = 13$, 6 [WT-FLX], $n = 10$, 5 [KO-V], and $n = 15$, 5 [KO-FLX], *$p < 0.05$, **$p < 0.01$, two-way RM-ANOVA with Bonferroni test). Error bars represent the SEM.

**Fluoxetine distinctly affects LTP and LTP-related mechanisms in WT and *Lrfn3*$^{-/-}$ mice.** The fluoxetine-dependent normalization of NMDAR hyperfunction and enhanced fear memory consolidation in *Lrfn3*$^{-/-}$ mice may involve NMDAR-dependent LTP in the amygdala and hippocampus, known to be associated with contextual fear memory acquisition/consolidation[57–64]. Alternatively, the normalization may involve changes in cellular or molecular events downstream of NMDAR activation that are

independent of LTP such as posttranslational protein modification or gene expression.

To test this, we first determined whether chronic fluoxetine treatment induces any changes in LTP in WT or *Lrfn3*$^{-/-}$ mice. HFS-LTP magnitudes at hippocampal SC-CA1 synapses were not different between vehicle-treated WT and *Lrfn3*$^{-/-}$ mice (Fig. 7a–d), consistent with the lack of a difference in HFS-LTP between naive WT and *Lrfn3*$^{-/-}$ mice (Supplementary Fig. 3).

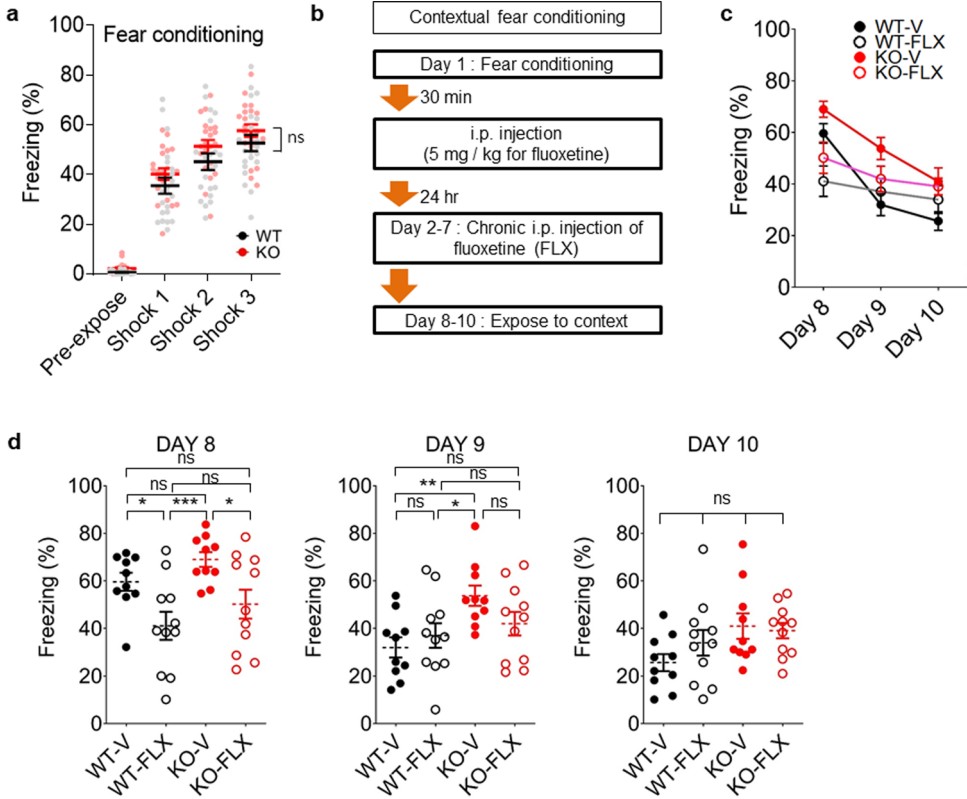

**Fig. 6 Fluoxetine normalizes enhanced fear memory consolidation in *Lrfn3*$^{-/-}$ mice. a** Normal contextual fear acquisition in *Lrfn3*$^{-/-}$ mice (8–12 weeks) in context A with foot-shock stimulation (0.8 mA) on day 1. ($n = 21$ mice [WT] and 21 [KO]; ns, not significant [genotype], two-way RM-ANOVA). **b** Experimental scheme for chronic fluoxetine/FLX treatment (5 mg/kg/d; 7 days; i.p.) of *Lrfn3*$^{-/-}$ mice (8–12 weeks) for the rescue of enhanced fear memory consolidation. After initial contextual fear memory acquisition (**a**), WT and KO mice were divided into four groups (WT-V/FLX and KO-V/FLX) for chronic drug treatments. **c** Summary of the quantification of the results from **b** during days 8–10. ($n = 10$ [WT-V], 11 [WT-FLX], 10 [KO-V], and 11 [KO-FLX]). **d** Subsets of the results displayed in **c** for days 8, 9, and 10. ($n = 10$ [WT-V], 11 [WT-FLX], 10 [KO-V], and 11 [KO-FLX], *$p < 0.05$, **$p < 0.01$, ***$p < 0.001$; ns, not significant, two-way RM-ANOVA with Bonferroni test).

Chronic fluoxetine treatment (5 mg/kg/d; 7 days; i.p.), however, induced a significant difference between HFS-LTP magnitudes between *Lrfn3*$^{-/-}$ and WT mice (Fig. 7a–g). This difference was caused by a trend towards decreased LTP in fluoxetine-treated WT mice and increased LTP in *Lrfn3*$^{-/-}$ mice; the results from WT mice are in line with a previous report that chronic (4 weeks) fluoxetine treatment induces LTP impairment[65], which likely occurs through decreased calcium flux involving increased ratios of GluN2A/GluN2B and GluA2/GluA1[65,66].

To better understand how chronic fluoxetine treatment induces distinct changes in LTP in WT and *Lrfn3*$^{-/-}$ mice, we measured levels of Ser-831- and Ser-845-phosphorylated AMPAR GluA1 subunits, which regulate single-channel properties and synaptic/surface targeting of GluA1, as well as synaptic plasticity[67–70], and are also known to be regulated by chronic fluoxetine[71,72]. Chronic fluoxetine treatment induced strong (>2-folds) increases in GluA1 Ser-831 and Ser-845 phosphorylation in *Lrfn3*$^{-/-}$ mice but not in WT mice, without affecting total protein levels (Fig. 8a, b and Supplementary Fig. 7).

Given that the function GluN2B-containing NMDARs was increased in *Lrfn3*$^{-/-}$ mice, and that GluN2B phosphorylation regulates aspects of NMDAR trafficking and function[73,74], we examined GluN2B phosphorylation at several sites (Ser-1303, Ser-1284, Tyr-1336, Tyr-1472, and Tyr-1480). Chronic fluoxetine treatment induced a significant difference in GluN2B phosphorylation at Ser-1303, but not at Ser-1284, Tyr-1336, Tyr-1472, or Tyr-1480, in the hippocampus (crude synaptosomes) of WT and *Lrfn3*$^{-/-}$ mice, without affecting total protein levels. The increase

in GluN2B Ser-1303 phosphorylation, which is mediated by CaMKIIα, is in line with the increased GluA1 Ser-831 phosphorylation, also known to be mediated by CaMKIIαβ[73,74], although less understood phosphatases may also be involved[75].

Taken together, these results suggest that chronic fluoxetine treatment has distinct effects on LTP and LTP-related mechanisms in WT and *Lrfn3*$^{-/-}$ mice. Given that fluoxetine does not significantly alter LTP in WT or *Lrfn3*$^{-/-}$ mice, it is less likely that LTP modulation underlies the fluoxetine-dependent normalization of contextual fear memory consolidation in *Lrfn3*$^{-/-}$ mice.

**Enhanced fear memory consolidation in *Lrfn3*$^{-/-}$ mice does not likely involve SALM3.** Lastly, given that SALM4 inhibits SALM2, SALM3, and SALM5[20], we sought to determine which SALMs (SALM2/3/5) might primarily contribute to the SALM4 deletion-induced NMDAR and fear memory phenotypes in *Lrfn3*$^{-/-}$ mice. To this end, we crossed *Lrfn3*$^{-/-}$ mice with *Lrfn4*$^{-/-}$ mice (lacking SALM3)[28] to achieve a double-KO (dKO) of SALM3 and SALM4, and tested whether the enhanced fear memory consolidation observed in *Lrfn3*$^{-/-}$ mice was normalized in *Lrfn3;Lrfn4*-dKO mice.

An analysis of the resulting dKO mice indicated that, although mice with single KO of SALM4 (*Lrfn3*$^{-/-}$ mice) showed increased fear memory consolidation at 48 h after acquisition, this phenotype was not normalized in *Lrfn3;Lrfn4*-dKO mice (Supplementary Fig. 8). These results suggest that SALM4 KO is less likely to involve SALM3, to induce abnormally enhanced fear memory consolidation (but see the results below). We could not test dKO of SALM4 with

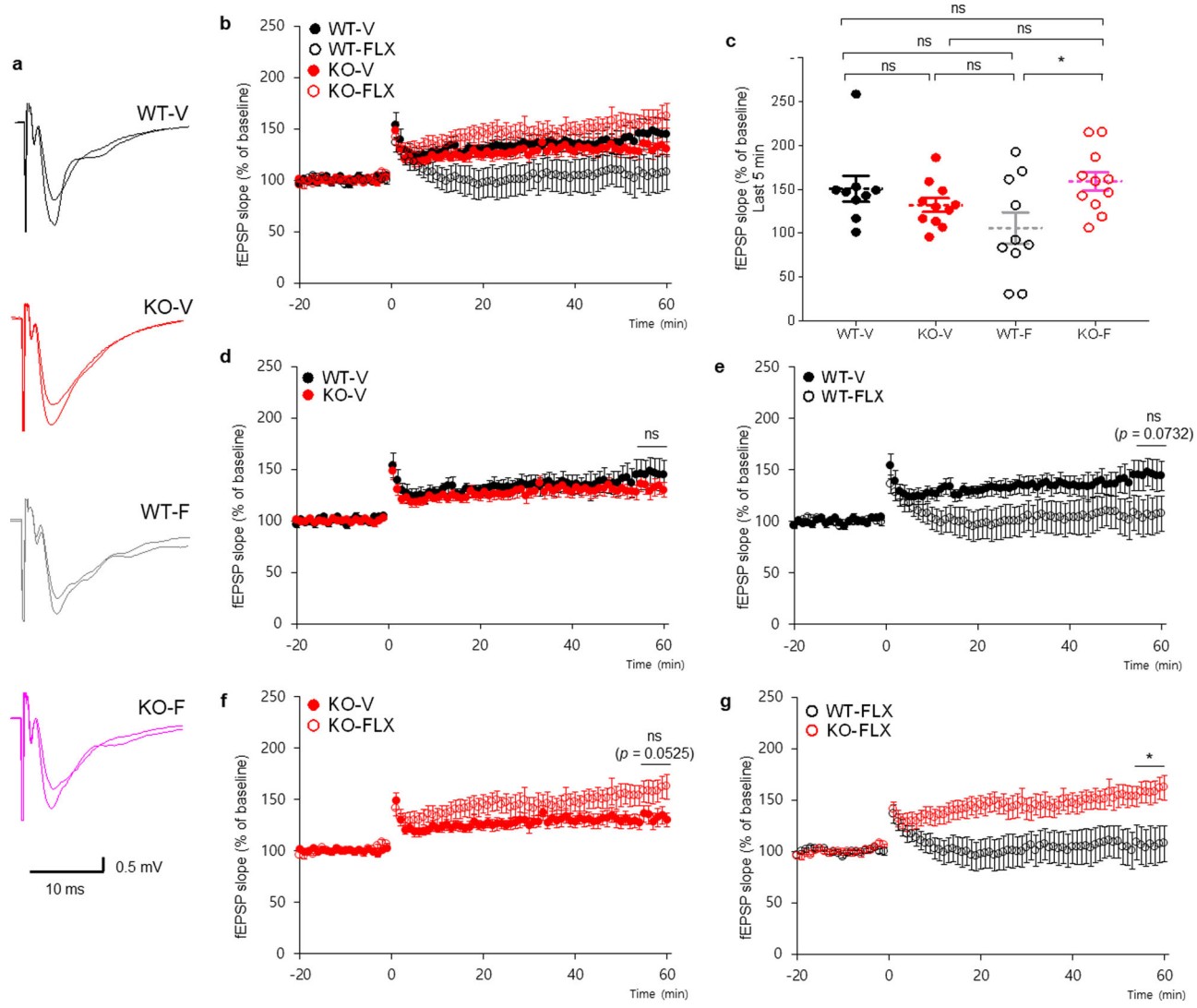

**Fig. 7 Distinct fluoxetine-induced changes of LTP in the WT and *Lrfn3*⁻/⁻ hippocampus. a** Sample traces for HFS-LTP at hippocampal SC-CA1 synapses in WT and *Lrfn3*⁻/⁻ mice (2–3 months) chronically treated with vehicle and fluoxetine (5 mg/kg/d; 7 days; i.p.). **b** Summary of the results from **a**. ($n = 9$ slices from 3 mice [WT-V/vehicle], 10, 3 [WT-FLX/fluoxetine], 11, 3 [KO-V], and 11, 3 [KO-FLX]). **c** Quantification of the results from **b**. ($n = 9$, 3 [WT-V/vehicle], 10, 3 [WT-FLX/fluoxetine], 11, 3 [KO-V], and 11, 3 [KO-FLX], *$p < 0.05$; ns, not significant, two-way RM-ANOVA with Tukey's test [last 5 min]). **d** A subset of results displayed in **b** comparing WT-V and KO-V. ($n = 9$, 3 [WT-V] and 11, 3 [KO-V], Student's *t*-test [last 5 min]). **e** A subset of results displayed in **b** comparing WT-V and WT-FLX. ($n = 9$, 3 [WT-V] and 10, 3 [WT-FLX], Student's *t*-test [last 5 min]). **f** A subset of results displayed in **b** comparing KO-V and KO-FLX. ($n = 11$, 3 [KO-V] and 11, 3 [KO-FLX], Student's *t*-test [last 5 min]). **g** A subset of results displayed in **b** comparing WT-FLX and KO-FLX. ($n = 10$, 3 [WT-FLX] and 11, 3 [KO-FLX], Student's *t*-test [last 5 min]). Error bars represent the SEM.

SALM2 or SALM5, because SALM2/5-KO mice were not available. In addition, we did not attempt an additional analysis of *Lrfn3;Lrfn4*-dKO mice (i.e., biochemical or synaptic measurements) because of the lack of the rescue effect in behavior.

**Presynaptic PTPσ is required for SALM4 deletion-induced NMDAR hyperactivity.** Postsynaptic SALM3 and SALM5 that are inhibited by SALM4[20] interact with presynaptic LAR-RPTPs, including PTPσ (encoded by *Ptprs*), to promote excitatory synapse development[28–30]. In addition, presynaptic LAR-RPTPs regulate postsynaptic NMDARs through mechanisms that are not yet clear[40,41]. It is thus possible that the loss of SALM4 may disinhibit SALM3 or SALM5 and promote their interactions with presynaptic LAR-RPTPs and subsequently induce NMDAR hyperactivity. Although the abovementioned results indicate that the dKO of SALM4 and SALM3 does not rescue the fear memory phenotype of *Lrfn3*⁻/⁻ mice, SALM5 remains intact in *Lrfn3*⁻/⁻ mice.

We thus attempted an acute knockdown of PTPσ in the hippocampal CA3 region (presynaptic to CA1), which has been shown to suppress NMDAR currents in the CA1 region[41], to see if this could rescue the SALM4 deletion-induced postsynaptic NMDAR hyperactivity in the CA1 region. The results indicated that PTPσ knockdown rescued NMDAR but not AMPAR responses in the CA1 region of the *Lrfn3*⁻/⁻ hippocampus (Fig. 9a–g and Supplementary Fig. 9). In contrast, WT synapses were not affected for NMDAR or AMPAR responses. These results suggest that presynaptic PTPσ is required for SALM4 deletion-induced NMDAR hyperactivity.

**Discussion**
In the present study, we demonstrated that SALM4 negatively regulates the function of NMDARs, but not AMPARs, through GluN2B, and that SALM4 contributes to the maintenance of normal contextual fear memory consolidation but not acquisition.

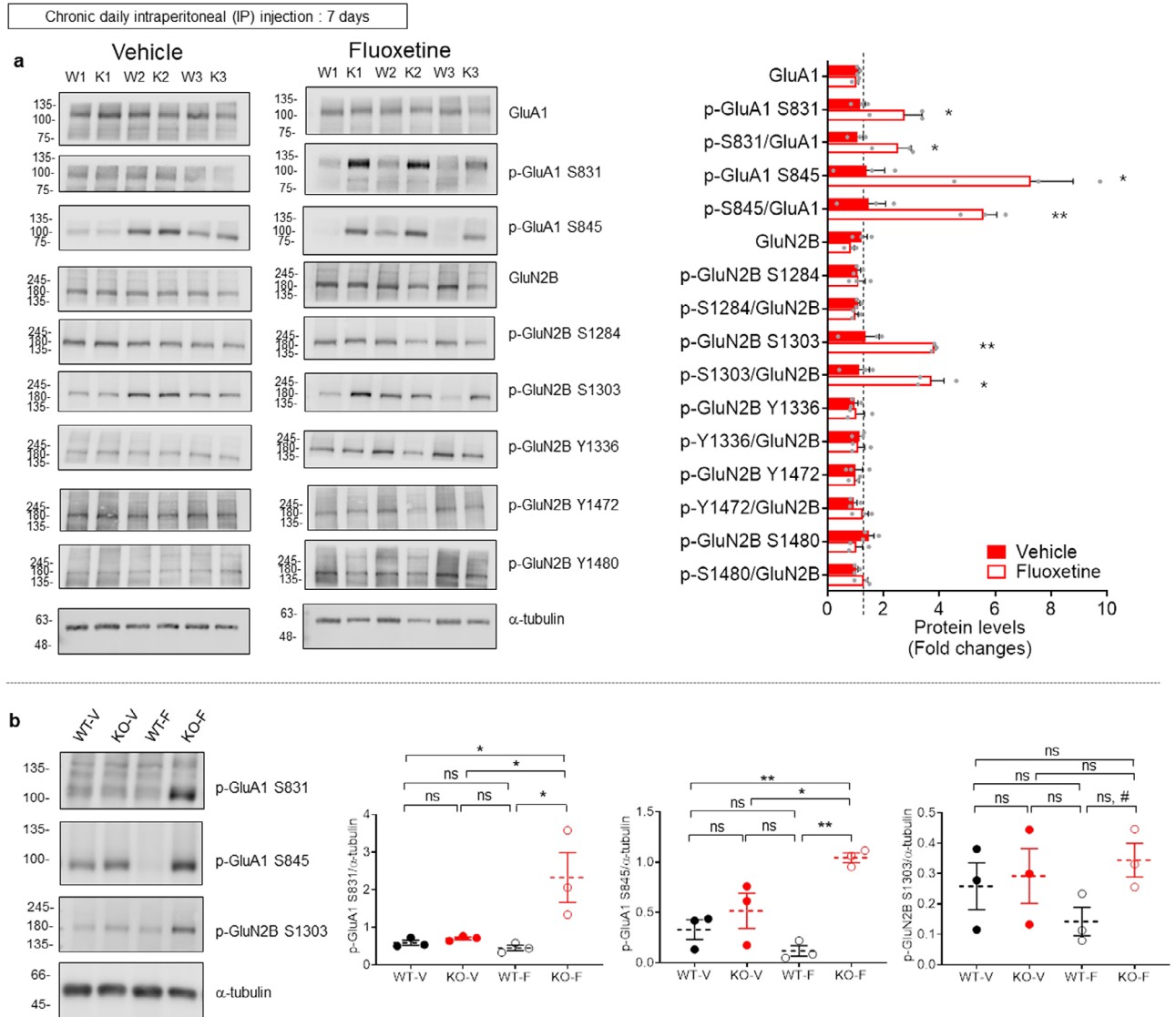

**Fig. 8 Distinct fluoxetine-induced changes in GluA1 and GluN2B phosphorylation in the WT and *Lrfn3*$^{-/-}$ hippocampus. a** Levels of GluA1 and GluN2B phosphorylation at different amino acid residues in the hippocampus (crude synaptosomes) of WT and *Lrfn3*$^{-/-}$ mice (2–3 months) chronically treated with vehicle or fluoxetine (5 mg/kg/d; 7 days; i.p.). It is noteworthy that fluoxetine treatment increased GluA1 Ser-831/845 and GluN2B S1303 phosphorylation without affecting total protein levels. Immunoblot signals from WT and KO mice normalized to α-tubulin signals were used to obtain KO/WT ratio values. ($n = 3$ mice [WT-V/vehicle], 3 [WT-F/fluoxetine], 3 [KO-V], and 3 [KO-F], *$p < 0.05$, **$p < 0.01$, Student's *t*-test). **b** Levels of GluA1 S831/S845 and GluN2B S1303 phosphorylation in chronic vehicle/fluoxetine-treated WT and *Lrfn3*$^{-/-}$ mice. All four samples were run on the same gel for two-way ANOVA comparisons. ($n = 3$ mice [WT-V/vehicle], 3 [WT-F/fluoxetine], 3 [KO-V], and 3 [KO-F], *$p < 0.05$, **$p < 0.01$; ns, not significant, two-way RM-ANOVA with Tukey's test (except for WT-F and KO-F in p-GluN2B S1303 where Student's *t*-test was used to obtain #$P < 0.05$)). Error bars represent the SEM.

In addition, we showed that chronic fluoxetine treatment rescues enhanced fear memory consolidation through mechanisms including the suppression of NMDAR function but not LTP.

*Lrfn3*$^{-/-}$ mice showed persistently increased NMDAR function, mainly involving GluN2B, in the hippocampus at both juvenile and adult stages (Fig. 2a–c). A quantitative analysis indicates that the increase in GluN2B-NMDAR currents is greater than the decrease in GluN2A-NMDAR currents by a factor of five, suggesting that the increase in GluN2B-NMDAR currents plays major roles in enhancing contextual fear memory consolidation, although we cannot exclude the possibility that the decreased GluN2A-NMDAR currents may also play a role.

Insight into possible mechanisms underlying these changes comes from a previous report that SALM4 associates with and inhibits SALM2-dependent excitatory synapse development[20]. SALM2 forms complexes in vivo with SALM1 and SALM3 in the brain[76], consistent with a recent crystallographic study showing that SALMs can form homo- and heterodimeric complexes[31–34]. SALM2 also associates with both NMDARs and AMPARs to promote excitatory synapse development[23], whereas SALM1 preferentially associates with and promotes dendritic clustering of NMDARs[22].

Therefore, SALM4 deletion may disinhibit SALM2 and SALM2-associated SALM1 and SALM3, where SALM1 is more likely to contribute to the NMDAR- but not AMPAR-specific hyperfunction observed in *Lrfn3*$^{-/-}$ mice. This hypothesis, however, is not supported by two recent independent studies on *Lrfn2*$^{-/-}$ mice lacking SALM1, which display either unaltered

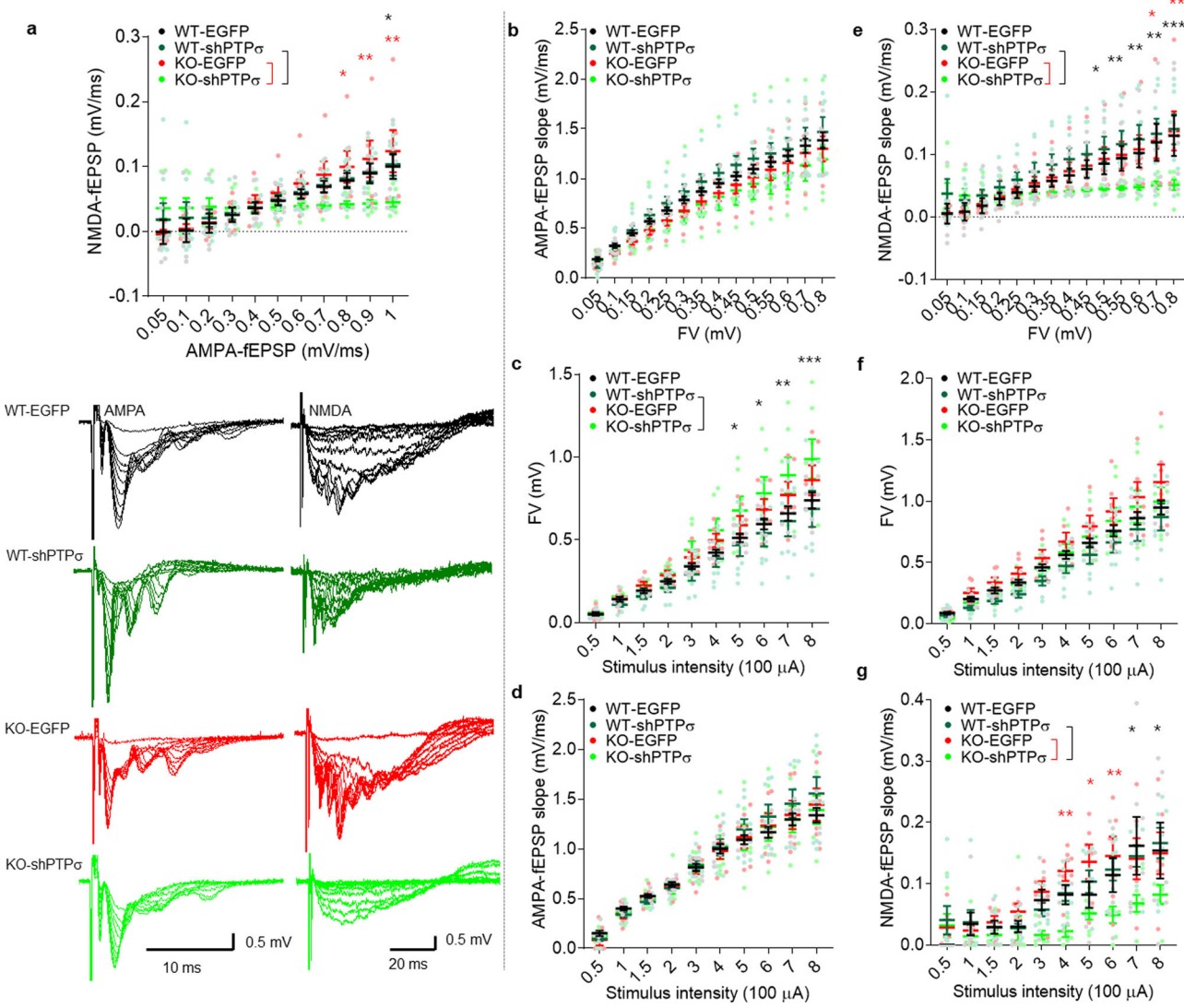

**Fig. 9 Presynaptic PTPσ is required for SALM4 deletion-induced NMDAR hyperactivity. a** Acute knockdown of PTPσ for 2 weeks in the CA3 region of the hippocampus in *Lrfn3*−/−, but not WT, mice (2–6 months) rescues NMDAR-fEPSPs but not AMPAR-fEPSPs at SC-CA1 synapses, as indicated by NMDAR-fEPSP slopes plotted against AMPAR-fEPSP slopes. (n = 7 slices from 3 mice [WT-EGFP/control], 8, 3 [WT-shPTPσ], 6, 3 [KO-EGFP], and 7, 3 [KO-shPTPσ], *p < 0.05, **p < 0.01, two-way RM-ANOVA with Bonferroni test). **b** PTPσ knockdown in WT or *Lrfn3*−/− CA3 neurons (2–6 months) does not affect AMPAR-fEPSPs at SC-CA1 synapses, as indicated by AMPAR-fEPSP slopes plotted against fiber volley amplitudes. (n = 7, 3 [WT-EGFP], 8, 3 [WT-shPTPσ], 6, 3 [KO-EGFP], and 7, 3 [KO-shPTPσ], two-way RM-ANOVA). **c** PTPσ knockdown in WT or *Lrfn3*−/− CA3 neurons (2–6 months) does not affect fiber volley amplitudes plotted against stimulation intensities under the context of AMPAR-fEPSPs measurements. (n = 7, 3 [WT-EGFP], 8, 3 [WT-shPTPσ], 6, 3 [KO-EGFP], and 7, 3 [KO-shPTPσ], *p < 0.05, **p < 0.01, ***p < 0.001, two-way RM-ANOVA with Bonferroni test). **d** PTPσ knockdown in WT or *Lrfn3*−/− CA3 neurons (2–6 months) does not affect AMPAR-fEPSPs at SC-CA1 synapses, as indicated by AMPAR-fEPSP slopes plotted against stimulation intensities. (n = 7, 3 [WT-EGFP], 8, 3 [WT-shPTPσ], 6, 3 [KO-EGFP], and 7, 3 [KO-shPTPσ], two-way RM-ANOVA). **e** PTPσ knockdown in WT or *Lrfn3*−/− CA3 neurons (2–6 months) rescues NMDAR-fEPSPs at SC-CA1 synapses, as indicated by NMDAR-fEPSP slopes plotted against fiber volley amplitudes. (n = 7, 3 [WT-EGFP], 8, 3 [WT-shPTPσ], 6, 3 [KO-EGFP], and 7, 3 [KO-shPTPσ], *p < 0.05, **p < 0.01, ***p < 0.001, two-way RM-ANOVA with Bonferroni test). **f** PTPσ knockdown in WT or *Lrfn3*−/− CA3 neurons (2–6 months) does not affect fiber volley amplitudes plotted against stimulation intensities under the context of NMDAR-fEPSPs measurements. (n = 7, 3 [WT-EGFP], 8, 3 [WT-shPTPσ], 6, 3 [KO-EGFP], and 7, 3 [KO-shPTPσ], two-way RM-ANOVA). **g** PTPσ knockdown in WT or *Lrfn3*−/− CA3 neurons (2–6 months) rescues NMDAR-fEPSPs at SC-CA1 synapses, as indicated by NMDAR-fEPSP slopes plotted against stimulation intensities. (n = 7, 3 [WT-EGFP], 8, 3 [WT-shPTPσ], 6, 3 [KO-EGFP], and 7, 3 [KO-shPTPσ], *p < 0.05, **p < 0.01, two-way RM-ANOVA with Bonferroni test). Error bars represent the SEM.

or increased NMDAR currents in the hippocampus[77,78]. In addition, the possibility of SALM3 disinhibition is not supported by the previous report in which SALM3 deletion in mice does not affect NMDAR function or NMDAR-dependent synaptic plasticity (TBS-LTP and LFS-LTD)[28]. More directly, our results indicate that dKO of SALM4 and SALM3 in mice does not normalize the enhanced fear memory consolidation in *Lrfn3*−/− mice (Supplementary Fig. 6). Lastly, whether SALM2

plays a role could not be tested, owing to the lack of reported SALM2-KO mice.

Alternatively, SALM4 deletion may promote NMDAR function through the modulation of the presynaptic adhesion partners of SALM3 or SALM5. Postsynaptic SALM4 interacts in *cis* with SALM3 and SALM5, and inhibits their *trans*-synaptic interactions with LAR-RPTPs, known to promote SALM3/5-dependent presynaptic differentiation[20,28–30]. Presynaptic LAR-RPTPs have

recently been shown to control postsynaptic NMDAR responses, but not AMPAR responses, through mechanisms that are not clearly defined[40,41]. In line with these results, presynaptic neurexins, a group of presynaptic adhesion molecules distinct from LAR-RPTPs, have been shown to regulate postsynaptic receptor responses and plasticity[36–38]. It is, therefore, possible that SALM4 deletion, through disinhibition of *trans*-synaptic interactions of SALM3/5 with presynaptic LAR-RPTPs and as yet unknown pre-to-post *trans*-synaptic mechanisms, promotes NMDAR responses, although SALM3 KO in mice does not affect NMDAR function[28]. Alternatively, SALM5 might play a role, although this hypothesis could not be tested for the lack of SALM5-KO mice in the present study. However, acute knockdown of PTPσ in the CA3 region of the *Lrfn3*[−/−] hippocampus rescues NMDAR function at SC-CA1 synapses (Fig. 9), suggesting the possibility that SALM4 deletion induces disinhibition of SALM5 and promotes SALM5- and PTPσ-dependent NMDAR hyperactivity (Supplementary Fig. 10). In support of this possibility, the GluN2B subunit of NMDARs is strongly decreased in PTPσ-mutant mice, as determined by immunoblot analysis of synaptic fractions and decay kinetics of NMDAR currents[41], similar to the stronger GluN2B component in *Lrfn3*[−/−] mice in the present study. Although further details remain to be determined, to the best of our knowledge, this study is the first to show that synaptic adhesion molecules can negatively regulate NMDAR responses. The only related studies are previous reports that MDGAs negatively regulate neuroligin-2-dependent inhibitory synapse development and neuroligin-1-dependent excitatory synapse development and postsynaptic AMPAR responses[16–21].

Behaviorally, *Lrfn3*[−/−] mice showed enhanced contextual fear memory consolidation but normal contextual fear memory acquisition, cued fear memory, spatial Memory, and object-recognition Memory (Fig. 1). Intriguingly, chronic treatment of *Lrfn3*[−/−] mice with fluoxetine, a medication used to treat excessive fear memory[51–53] that can inhibit GluN2B-containing, but not GluN2A-containing, NMDARs[54,55], normalized NMDAR hyperfunction and enhanced fear memory consolidation in *Lrfn3*[−/−] mice (Figs. 5 and 6). However, ifenprodil (a selective GluN2B antagonist) did not rescue the fear phenotype in *Lrfn3*[−/−] mice (Supplementary Fig. 6). This suggests that NMDAR inhibition may not be the main determinant for the fluoxetine-dependent behavioral rescue, although there was a tendency for the rescue of enhanced fear memory consolidation, and the ifenprodil-dependent rescue might need optimization.

Another interesting finding in this study is that NMDAR hyperfunction in *Lrfn3*[−/−] mice promotes contextual fear memory consolidation but not acquisition. Previous studies have shown that LTP is critical for contextual fear memory conditioning/acquisition[57–60]. Therefore, the fact that *Lrfn3*[−/−] mice showed normal LTP (Supplementary Fig. 3), but displayed increased NMDAR currents, may explain the normal contextual acquisition in *Lrfn3*[−/−] mice and points to an interesting situation in which contextual fear conditioning and consolidation may involve different NMDAR-related mechanisms (LTP vs. other cell biological/molecular events downstream of NMDAR activation), at least in *Lrfn3*[−/−] mice. This idea is further supported by the observation that chronic fluoxetine treatment, which rescued the enhanced contextual fear consolidation in *Lrfn3*[−/−] mice (Fig. 6), normalized NMDAR hyperfunction (Fig. 5) through GluN2B/GluA1 hyperphosphorylation (Figs. 7 and 8), without affecting LTP. Possible cell biological/molecular events downstream of GluN2B-NMDAR activation independent of LTP modulation in *Lrfn3*[−/−] mice include posttranslational modification of synaptic/neuronal proteins or changes in gene expression.

In conclusion, the current study demonstrates that SALM4 negatively regulates NMDAR, but not AMPAR, function through GluN2B and suppresses excessive fear memory consolidation, but not acquisition, through mechanisms, including NMDAR hyperfunction independent of LTP modulation.

## Methods

**Animals**. *Lrfn3*[−/−] animals with SALM4 deletion have been described[20]. The experimental procedures described below were evaluated and approved by the Committee on Animal Research at Korea Advanced Institute of Science and Technology (KAIST; approval numbers KA2020-32 and KA2020-96). Mice were housed and bred in the mouse facility of KAIST, and were maintained according to the Animal Research Requirements of KAIST, being housed and fed ad libitum under 12 h light/dark cycle at 21 °C and 50–60% humidity. All behavioral experiments were performed during dark-cycle periods. Only male mice were used for behavioral and all other experiments.

**Cued fear conditioning**. A mouse was introduced to the fear conditioning box (Coulbourn Instrument). On the first day, a mouse was placed in the fear box (context A) for 5 min for habituation and to check basal freezing levels. On the conditioning day, 24 h later, a mouse was introduced to the same fear box and then given 3 min to explore the environment, followed by three foot shocks (0.8 mA, 1 s, unconditioned stimulus (US)) at the end of sound (75 dB, 8 kHz tone, 20 s, conditioned stimulus (CS)) with 1 min intervals for 3 min (this process totally take 6 min). On the day after, the mice that already cued fear-conditioned were introduced to a different fear box (context B) and allowed to move for 3 min without tone (CS−) and 3 min with tone (CS+; 75 dB, 8 kHz tone, 3 min). After 24 h, the mice were re-introduced to the original fear box (context A) and allowed to move for 6 min. Freezing behaviors were analyzed using FreezeFrame 3 (Coulbourn Instrument).

**Contextual fear conditioning**. A mouse was introduced into the fear-conditioning box on day 1 for 5 m for habituation and to evaluate basal freezing levels. On the test day (24 h later), the mouse was introduced into the same fear-conditioning box and then given 2 m to explore the environment, followed by three foot shocks (0.8 mA, 1 s, US) at 1 min intervals for 3 m. On the day after, mice that had already received foot shocks were re-introduced into the same fear-conditioning box and allowed to move freely for 5 m for short/long-term memory and for 6 m for extinction to measure fear levels. Freezing behaviors were analyzed using FreezeFrame 3.

For chronic fluoxetine treatment, fluoxetine (fluoxetine hydrochloride; Sigma-Aldrich) was dissolved in saline and injected i.p. into mice at a dose of 5 mg/kg/d. For chronic DCS treatment, DCS (Sigma-Aldrich) was dissolved in saline and injected i.p. into mice at a dose of 20 mg/kg/d. For chronic memantine treatment, memantine (Sigma-Aldrich) was dissolved in saline and injected i.p. into mice at a dose of 10 mg/kg/d. For chronic ifenprodil treatment, Ifenprodil (+)-tartrate salt (Sigma-Aldrich) was dissolved in saline and injected i.p. into mice at a dose of 10 mg/kg/d. Control mice were injected i.p. with the same volume of saline. Injections were administered 30 min after contextual fear conditioning, followed by seven daily injections before re-exposure to a fear-conditioning context.

**Morris water maze**. Mice were trained to find the hidden platform (10 cm diameter) in a white plastic tank (23 °C, 120 cm diameter, ~120 lux). Mice were given four trials per day with an inter-trial interval of 1 h. Training experiments were performed for 7 consecutive days and the probe tests were given for 1 min with the platform removed from the pool at day 8 (24 h after the last training session). Reverse training experiments were performed with the platform location changed for 4 consecutive days and the probe test was given for 1 min with the platform removed at day 5 (day 13 from the start day of the Morris water maze test). Percentage of time spent in four quadrants of the pool (T, target; O, opposite; L, left; R, right), the number of exact crossings over the platform area, and swimming speed were analyzed using Ethovision 3.1 program (Noldus).

**Novel object recognition**. The novel object-recognition test was performed in an open-field apparatus (200 lux). This test consisted of three steps: a mouse was (1) allowed to freely move in the open-field box for habituation for 60 min; (2) allowed to explore two identical objects for 10 min, sample phase; and (3) 1 day after, one of the two objects was replaced with a new one and the mouse was allowed to explore two objects for 10 min, test phase. An interest to "familiar object" and "novel object" was scored when the nose of a mouse was in contact with the object or directed toward the object within the region <2 cm from the object, except for when the mouse was on top of the object with all four feet.

**Electrophysiology**. Recordings were made using MultiClamp 700B amplifier (Molecular Devices) and Digidata 1440 A (Molecular Devices) were used for electrophysiological experiments. Data were acquired and analyzed by Clampex 10.2 (Molecular Devices). Drugs were purchased from Tocris (NBQX, D-AP5) and Sigma (picrotoxin, glycine) (see Table 1 for details).

**Table 1 Key reagents and resources used in the present study.**

| Reagent or resource | Source | Identifier |
|---|---|---|
| Antibodies | | |
| Guinea pig polyclonal anti-PTPσ | Made in-house | #2135 (1:1000) |
| Rabbit polyclonal anti-Homer | Made in-house | #1133 (1:1000) |
| Rabbit polyclonal anti-GluA1 | Made in-house | #1193 (1:1000) |
| Rabbit polyclonal anti-phospho-GluA1 Ser-831 | Millipore | #AB5847 (1:500) |
| Rabbit polyclonal anti-phospho-GluA1 Ser-845 | Millipore | #AB5849 (1:500) |
| Rabbit polyclonal anti-GluA2 | Made in-house | #1195 (1:1000) |
| Rabbit polyclonal anti-phospho-GluA2 (Tyr869/Tyr873/Tyr876) | CST | #3921 (1:500) |
| Guinea pig polyclonal anti-CamKII | Made in-house | #1300 (1:1000) |
| Rabbit polyclonal anti-phospho CamKII (T286) | Abcam | #ab32678 (1:1000) |
| Rabbit polyclonal anti-PKAα + β (catalytic subunits-N term) | Abcam | #ab71764 (1:1000) |
| Mouse monoclonal anti-PKCα | BD Biosciences | #610108 (1:1000) |
| Rabbit polyclonal anti-phospho-PKC (pan) (βII Ser660) | CST | #9371 (1:500) |
| Rabbit polyclonal anti-p44/42 MAPK (Erk1/2) | CST | #9102 (1:500) |
| Rabbit polyclonal anti-phospho-p44/42 MAPK (Erk1/2) (Thr202/Tyr204) | CST | #9101 (1:500) |
| Rabbit monoclonal anti-CREB (48H2) | CST | #9197 (1:500) |
| Rabbit monoclonal anti-phospho-CREB (Ser133) (87G3) | CST | #9198 (1:500) |
| Rabbit monoclonal anti-GSK3β (27C10) | CST | #9315 (1:500) |
| Rabbit polyclonal anti-phospho-GSK3β (Ser9) | CST | #9336 (1:500) |
| Rabbit polyclonal anti-phospho-GluN2B (Tyr-1472) | CST | #4208 (1:500) |
| Rabbit polyclonal anti-phospho-GluN2B (Ser-1284) | CST | #5355 (1:500) |
| Rabbit polyclonal anti-phospho-GluN2B (Tyr-1472) | CST | #4208 (1:500) |
| Rabbit polyclonal anti-phospho-GluN2B (Tyr-1336) | Abcam | #ab138664 (1:500) |
| Rabbit polyclonal anti-phospho-GluN2B (Ser1480) | Abcam | #ab73014 (1:500) |
| Rabbit polyclonal anti-phospho-GluN2B (Ser-1303) | Milipore | #07-398 (1:500) |
| Rabbit polyclonal anti-GluN2B (extracellular) | Almone Labs | #AGC-003 (1:1000) |
| Rabbit polyclonal anti-GluN2A | Milipore | #07-632 (1:1000) |
| Mouse monoclonal anti-GluN1/NR1 | Neuromab | #75-272 (1:500) |
| Mouse monoclonal anti-β-catenin | BD Transduction | #610154 (1:1000) |
| Mouse monoclonal anti-α-tubulin | DSHB | #12G10 (1:10,000) |
| Guinea pig polyclonal anti-EGFP | Made in-house | #1998 (1:1000) |
| Bacterial and virus strains | | |
| pAAV-U6-GFP | Cell Biolabs | Cat #VPK-413 |
| pAAV-U6-sh-PTPσ-GFP | Made in-house (Jaewon Ko) | NA |
| Chemicals, peptides, and recombinant proteins | | |
| D(+)-sucrose | PanReacAppliChem | Cat#131621.0914 |
| Sodium chloride | Sigma | Cat#S7653 |
| Sodium bicarbonate | Sigma | Cat#S6297 |
| D-(+)-glucose | Sigma | Cat#G7528 |
| Potassium chloride | Sigma | Cat#P3911 |
| Sodium pyrophosphate decahydrate | Sigma | Cat#221368 |
| Na-pyruvate | Sigma | Cat#P2256 |
| L-ascorbic acid (sodium salt) | Sigma | Cat#A4034 |
| Calcium chloride dehydrate | Sigma | Cat#C3881 |
| Magnesium chloride hexahydrate | Sigma | Cat#M0250 |
| Tetraethylammonium chloride (TTX) | Sigma | Cat#T2265 |
| Picrotoxin | Abcam | Cat#ab120315 |
| D-AP5 | Tocris | Cat#0106 |
| NBQX disodium salt | Tocris | Cat#1044 |
| Glycine | Panreac | Cat#A1067 |
| Ifenprodil (+)-tartrate salt | Sigma | Cat#I2892 |
| Cesium methanesulfonate | Sigma | Cat#C1426 |
| Tetraethylammonium chloride | Sigma | Cat#T2265 |
| HEPES | Sigma | Cat#H4034 |
| QX-314 chloride salt | Alomone | Cat#Q150 |
| Adenosine 5′-triphosphate magnesium salt | Sigma | Cat#A9187 |
| Guanosine 5′-triphosphate sodium salt hydrate | Sigma | Cat#G8877 |
| Ethylene glycol-bis(2-aminoethylether)-N,N,N′,N′-tetraacetic acid | Sigma | Cat#E3889 |
| D-cycloserine | Sigma | Cat#C6880 |
| Memantine hydrochloride | Sigma | Cat#M9292 |
| Fluoxetine hydrochloride | Sigma | Cat#F132 |
| Software and algorithms/others | | |
| LABORAS | METRIS | https://www.metris.nl/laboras/laboras.htm |

**Table 1 (continued)**

| Reagent or resource | Source | Identifier |
|---|---|---|
| FreezeFrame 3 | Coulbourn Instruments | https://www.coulbourn.com/category_s/277.htm |
| Ethovision XT 10.1 | Noldus | https://www.noldus.com/ |
| Clampfit 10.7 | Molecular devices | NA |
| pCLAMP 10.1 | Molecular devices | NA |
| Multiclamp Commander 700B | Molecular devices | NA |
| Digidata 1550 | Molecular devices | NA |
| LI-COR Odyssey (version) Fc | LI-COR Biosciences | https://www.licor.com/bio/odyssey-xf/ |
| Image Studio Lite Ver. 5.2.5 | LI-COR Biosciences | https://www.licor.com/bio/image-studio-lite/ |
| GraphPad Prism 8.0 | GraphPad | https://www.graphpad.com/ |
| LSM780 | Carl Zeiss | https://bio.kaist.ac.kr/index.php?mid=bio_facilities&document_srl=376 |
| TCS SP8 Dichroic/CS | Leica | https://www.leica-microsystems.com/fileadmin/pdf_uploads/koch-TCS-SP8_en_Brochure_FINAL.pdf |
| ImageJ | NIH | imagej.net |

Key reagents and resources used in the present study are shown with their sources (in-house or commercial) and related identifiers and antibody dilutions used.

For the NMDA/AMPA ratio, hippocampal slices (300 μm) were prepared and recovered at 32 °C for recovery before recording (25–27 °C). Pipette (2.5–3.5 MΩ) solutions contained (in mM) 117 CsMeSO$_4$, 8 NaCl, 10 TEACl, 10 HEPES, 5 Qx-314Cl, 4 Mg-ATP, 0.3 Na-GTP, and 10 EGTA (295 mOsm). Input resistance levels in recording electrodes and artificial cerebrospinal fluid (ACSF)-filled stimulator were 2.8–3 and 1.5–2 MΩ, respectively. Picrotoxin (100 μM) was added to oxygenated ACSF. CA1 pyramidal cells were held at −70 mV after baseline stabilization and stimulated at every 15 s (first 10 min, AMPA-mediated EPSCs, 30 consecutive responses). Subsequently, membrane potentials were switched to +40 mV, and after 10 min stabilization, NMDAR-EPSCs were evoked and the currents at 60 ms after the stimulation were used for analyses. The NMDA/AMPA ratio was determined by dividing the mean value of 30 NMDAR-EPSCs by the mean value of 30 AMPAR-EPSC peak amplitudes. Data were acquired and analyzed using Clampex 10.4 (Molecular Devices).

To measure GluN2B-containing NMDAR currents sensitive to ifenprodil, preparation of hippocampal slices and measurements of AMPAR- and NMDAR-EPSCs were performed as described above for the NMDA/AMPA ratio. To measure GluN2B-containing NMDAR currents, slices were incubated with oxygenated ACSF containing ifenprodil (3 μM) and picrotoxin (100 μM) starting from 5 min after the acquisition of stable baseline NMDA-EPSCs. The data were acquired and analyzed using Clampex 10.4 (Molecular Devices).

For field recordings, hippocampal slices (400 μm) were prepared from 8- to 9-week-old male littermates. The brain was rapidly isolated and placed to cold, oxygenated (95% O$_2$ and 5% CO$_2$) dissection buffer containing (in mM) 212 sucrose, 10 glucose, 25 NaHCO$_3$, 5 KCl, 1.25 NaH$_2$PO$_4$, 1.2 L-ascorbate, 2 pyruvate with 3.5 MgCl$_2$, and 0.5 CaCl$_2$. Hippocampal slices were prepared using Leica VT1000P vibratome (Leica) and transferred for recovery to a holding chamber containing oxygenated ACSF containing (in mM) 125 NaCl, 10 glucose, 25 NaHCO$_3$, 2.5 KCl, 1.25 NaH$_2$PO$_4$ with 1.3 MgCl$_2$, and 2.5 CaCl$_2$ at 32 °C for recovery, and slices were transferred to a recording chamber where they were perfused continuously with oxygenated ACSF (27–28 °C). Hippocampal CA1 fEPSPs were evoked by Schaffer collateral stimulation (0.2 ms current pulses) using a concentric bipolar electrode. Synaptic responses were recorded using ACSF-filled microelectrodes (1 MΩ), The three successive responses were averaged and expressed relative to the normalized baseline. TBS-LTP was induced by four episodes of TBS with 10 s intervals. TBS consisted of ten stimulus trains delivered at 5 Hz with each train consisted of four pulses at 100 Hz. For HFS-LTP, HFS (100 Hz, 1 s) was applied after a stable baseline was acquired and quantified using the initial slopes of fEPSPs. LTD was induced by SP-LFS (1 Hz for 900 s). HFS-LTP at TA-CA1 synapses was measured using the condition identical to that used for HFS-LTP at SC-CA1 synapses, except for the lack of picrotoxin and the stimulation of the TA pathway. Data from slices with stable recordings (<5% change over the baseline period) were included in the analysis. All data are presented as mean ± SEM normalized to the preconditioning baseline (at least 20 min of stable responses). Recordings were performed using an AM-1800 Microelectrode amplifier (A-M Systems), and IGOR software (Wavemetrics) was used for digitizing and analyzing the responses. All data are presented as mean ± SEM normalized to the preconditioning baseline (at least 20 min of stable responses).

AMPAR- and NMDAR-mediated fEPSPs were isolated by ACSF containing (in mM) 125 NaCl, 10 glucose, 25 NaHCO$_3$, 2.5 KCl, 1.25 NaH$_2$PO$_4$ with 2.5 CaCl$_2$, 1.3 MgCl$_2$, picrotoxin (100 μM), D-AP5 (50 μM) for AMPAR-fEPSPs, and with 2.5 CaCl$_2$, 0.1 MgCl$_2$, glycine (1 mM), picrotoxin (100 μM), NBQX (10 μM) for NMDAR-fEPSPs, respectively. The Schaffer collateral pathway was stimulated every 20 s and three responses were averaged. The stimulation gradually increased with 1 min interval. AMPAR-fEPSPs were recorded first in ACSF with AP5 and the solution was changed to that containing NBQX to isolate and measure NMDAR-fEPSPs in the same slices; NMDAR-fEPSPs were measured after at least 30 min washout of AP5 and stabilization of NMDAR-fEPSPs. The stimulation gradually increased with 1 min interval. The data were acquired, analyzed, and fitted with exponential, cumulative probability by Clampex 10.4 (Molecular Devices). In addition, rearranged data were artificially organized, based on the same criteria.

**Laboras test**. To monitor long-term mouse movements, mice in the cage of the Laboratory Animal Behavior Observation Registration and Analysis System (Laboras, Metris) with fresh bedding were recorded of their movements starting from 2 h prior to light-off phase for 72 h. The light was set at 70–90 lux, similar to their original home cage conditions. For recordings, four to six grouped mice were separated and individually caged in Laboras cages placed on top of a vibration-sensitive platform. The parameters analyzed were distance moved, climbing, grooming, and rearing. Recorded data were taken and calculated using LABORAS 2.6 software (Metris).

**Open-field test**. The open-field apparatus consisted of a square, white plastic open box (40 × 40 × 40 cm, 200 lux). The center zone lines were 10 cm apart from the edges of the box and mice were introduced to the center region of the apparatus at the start of the test. Mouse movements were recorded with a video camera for 60 min and analyzed using the Ethovision 3.1 program (Noldus).

**Rotarod test**. To perform this test, the rolling speed of the rod was constant at 14 r.p.m. or gradually increased from 4 to 40 r.p.m. within 5 min. Then, each mouse was placed on the rotating rod for a second, followed by the start of rod rolling. The test was performed for 3 consecutive days (constant 14 r.p.m. speed) and for 7 consecutive days, followed by 2 additional days and another 1 day (accelerating 4–40 r.p.m. speed), while recording the latencies of each mouse falling from the rod to the bottom.

**Three-chamber test**. The three-chamber test[79] was performed using the apparatus with the dimension of 40 cm (W) × 20 cm (D) × 25 cm (H) was used. A mouse was separated from their home cages to a freshly bedded new cage and left alone for 10 min before it was introduced to the middle chamber of the apparatus (30–40 lux). This test was composed of several steps as follows: a mouse was (1) allowed to freely move in the apparatus with empty wired cages on side chambers for 10 min for habituation, (2) guided and confined to center, (3) allowed to explore side chambers with wired cages, which contain either an inanimate object or a stranger mouse (termed stranger 1, C57BL/6J) for 10 min, (4) guided and

confined to center, (5) the object in the wired cage was replaced with a new stranger mouse (stranger 2), and (6) allowed to explore the two mice for 10 min. Exploration of "object," "stranger 1," or "stranger 2" was scored when the nose of the mouse was close (<1 cm) to or contacted the wired cage.

**Nesting behavior**. Two square nestlets were used as nest-building materials. A mouse was maintained in its home cage during the 4-day session and the status of the nesting material was photographed every day. Nest building was scored as follows: score 0 for no nesting, score 1 for a pad-shaped flat nest, score 2 for complex nests including biting and warping the tissue, score 3 for a cup-shaped nest, and score 4 for a hollow hemisphere with one opening.

**Prepulse inhibition**. Prepulse inhibition (PPI) of the acoustic startle response was measured with "Med pro startle response" equipment (Med Associated, Inc.). After 5 min of acclimation to the apparatus, mice were given eight different types of trials with "white-noise" stimulus type: trials with the startle stimulus only (40 ms 120 dB, 1 ms rise/fall time); trials with the prepulse stimuli (20 ms, 1 ms rise/fall time) that were 4, 12, or 20 dB above the white-noise background (70 dB) and followed 100 ms later with the startle stimulus; and trials with background stimuli (null trials) to control for background movements of the animals. Total stimulus sample duration was 500 ms and the null period was 200 ms. Test sessions were composed of 48 trials (8 trials × 6 blocks), followed by each block of 1 null, 1 startle trial, 3 prepulse, and 3 prepulse–startle trials presented in a pseudorandom order, and ending with 1 null trial (250 ms) for checking the baseline movement after test session. Total test time takes 10 min, including acclimation time to reduce the stressful condition of mice. The average inter-trial interval was 15 s, with a range of 10–20 s. The largest peak startle response for each trial was measured with the following three criteria: (1) 20 ms of minimum latency, (2) minimum peak value 50, and (3) 30 ms of minimum peak time after the onset of the startle stimulus. PPI was calculated as %PPI = [1 − (prepulse + startle trials/startle-only trials)] × 100.

**Repetitive behaviors**. To measure grooming and rearing, a mouse was placed in an empty new home cage with a transparent acryl board lid for 10 min (0 lux). Grooming behavior was defined as stroking down or scratching the face, head, or body with two forelimbs observed during the last 5 min. Rearing was defined as raising of both forepaws during the last 5 min. For digging, a mouse was placed in a fresh-bedded home cage for 10 min (0 lux). Digging behavior was defined as the behavior of a mouse when it coordinately uses two forepaws or hind legs to displace bedding materials during the last 5 min. In the marble or food pellet burying test, a home cage was filled with bedding materials, ~5 cm deep. Glass marbles (1.5 cm in diameter) or food pellets (1.5 × 2–2.5 cm) was placed on the bed surface (4 × 5 arrangement). A mouse was introduced to the center and observed for 30 min. The numbers of buried (>2/3 depth) marbles or food pellets were counted.

**Elevated plus maze**. The elevated plus maze consisted of two closed arms (5–10 lux), two open arms (250 lux), and a center area (each arm 30 cm long, center 5 × 5 cm). This apparatus had a shape of a cross and was elevated to 50 cm from the floor. For experiments, a mouse was placed in the center area and allowed to explore the space for 10 min. Translocation from one compartment to another was scored when all four feet of the mouse move to the other area.

**Light–dark test**. The light–dark apparatus consisted of light (20 × 30 × 20 cm, 650–700 lux) and dark (20 × 13 × 20 cm, 0–20 lux) chambers adhered to each other. A mouse was placed in the center of the light chamber and allowed to move freely across the light and dark chambers through a gate for 10 min. Translocation was scored when all four feet of the mouse move to the other area.

**Tail suspension test**. The tail suspension test was performed at 90 cm height (300 lux). The tail of each mouse was attached to the top of the apparatus, with the mouse hanging upside down for 7 min. Before this test, the weight of each mouse was recorded to exclude the factor of gravity and there was no genotype difference.

**Forced swimming test**. The forced swimming test was performed during two sequential days. The apparatus, a 3 l pyrex-glass beaker (15 cm diameter), was filled with transparent water (23–25 °C, 15 cm depth, 180–200 lux). On the first day, a mouse was introduced to the apparatus for 15 min and mouse movements during the last 5 min were analyzed to determine baseline movements. On day 2, the mouse was re-introduced to the same environment for 5 min and mouse movements during the 5 min were analyzed to determine forced-swim levels.

**Fluorescence in situ hybridization**. Frozen mouse brain sections (14 μm thick) were cut coronally through the hippocampal formation and thaw-mounted onto Superfrost Plus Microscope Slides (Fisher Scientific). The sections were fixed in 4% paraformaldehyde, followed by dehydration in increasing concentrations of ethanol and protease digestion. For hybridization, the sections were incubated in different amplifier solutions in a HybEZ hybridization oven (Advanced Cell Diagnostics) at

40 °C. The probes used in these studies were three synthetic oligonucleotides complementary to the nucleotide sequence 2272–2692 of Mm-*Lrfn3*-C1, 62–3113 of Mm-*Gad1*-C2, 552–1506 of Mm-*Gad2*-C3, 464–1415 of Mm-*Slc17a7* (Vglut1)-C2, and 1986–2998 of Mm-*Slc17a6* (Vglut2)-C3 (Advanced Cell Diagnostics). The labeled probes were conjugated to Alexa Fluor 488, Altto 550, or Atto 647. The sections were hybridized with probe mixtures at 40 °C for 2 h. Nonspecifically hybridized probes were removed by washing the sections in 1× wash buffer and the slides were treated with Amplifier 1-FL for 30 min, Amplifier 2-FL for 15 min, Amplifier 3-FL for 30 min, and Amplifier 4 Alt B-FL for 15 min. Each amplifier was removed by washing with 1× wash buffer. The slides were viewed and photographed using TCS SP8 Dichroic/CS (Leica), followed by image analyses using ImageJ program, as described previously[80].

**PTPσ knockdown**. The short hairpin RNA (shRNA) adeno-associated virus (AAV) against the mouse *Ptprs* gene was constructed by annealing, phosphorylating, and cloning oligonucleotides targeting the mouse *Ptprs* gene (GenBank accession: XM_006523874.2; 5′-GCCACACACCTTCTATAAT-3′) into the BamHI and EcoRI sites of the pAAV-U6-GFP vector (Cell BioLabs). The shRNA sequence was previously validated[81]. The CA3 region of the hippocampus in WT or *Lrfn3*$^{-/-}$ mice (2–6 months) was injected with AAV-U6-shPTPσ-GFP and allowed to express the delivered constructs for 2 weeks as described previously[41]. AAV-U6-shPTPσ-GFP-dependent expression of shRNAs (control/empty and shPTPσ) was confirmed by positive GFP (green fluorescent protein) signals, and PTPσ knockdown was confirmed by immunoblot analysis of CA3 lysates.

**Statistics and reproducibility**. Statistical details, including mouse numbers, are described in Supplementary Data 2 file.

**Reporting summary**. Further information on research design is available in the Nature Research Reporting Summary linked to this article.

## Data availability

All data supporting the findings of the current study are provided in the paper and Supplementary Information. All source data are provided as Supplementary Data 1 with this paper. All additional information will be made available upon request to the authors.

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

## Acknowledgements

This work was supported by the National Research Foundation of Korea (NRF-2017M3C7A1079692 to H.K.) the Brain Research Program (2017M3C7A1023470 to J.K.) and the Institute for Basic Science (IBS-R002-D1 to E.K.).

## Author contributions

E.L. and Y.Y. performed behavioral experiments. E.L., E.-J.L., W.S., S.M.U. and T.-Y.C. performed electrophysiology experiments. E.L. performed immunoblot experiments. E.L and Y.Y. contributed to mouse genotyping and quantitative analyses. E.Y. performed FISH experiments. K.A.H. generated PTPσ knockdown vector. E.L., K.K., Y.Y., S.L. and M.B. performed PTPσ knockdown-related viral injection and characterization experiments. E.L., S.-Y.C., H.K., J.K. and E.K. designed research and wrote the manuscript.

## Competing interests

The authors declare no competing interests.
