## [Transparent Peer Review File · Communications Biology]

Reviewers' comments:

Reviewer #1 (Remarks to the Author):

This study addresses the synaptic role of the synaptic cell adhesion-like molecule SALM-4, using the corresponding KO mice. The enhanced GluN2B-mediated NMDAR function associated with enhanced contextual fear memory consolidation indicates that SALM4 participates in the suppression of excessive

GluNB-containing NMDAR function and in fear memory consolidation

This is an interesting piece of work describing how the SALM-4 KO affects fear memory, glutamate receptor activity and levels without (supposedly) affecting synaptic plasticity, at least LTP at SC to CA1 synapses. It is presented in an organized and clear manner. Moreover, effects of NMDAR agonists and antagonists were tested

In the discussion, some minor issues should be addressed:

- 1) Other forms of synaptic plasticity (LTD) or plasticity in other hippocampal pathways have not been examined. Given that the molecular composition of hippocampal synapses vary in a circuit-dependent manner (and even in a distal vs proximal location, well described at least in the stratum radiatum), it is additionally possible that the SC-CA1 pathway is spared from LTP deficiency while other pathways are affected. Moreover, the increase of GluN2B-mediated synaptic responses should enhance plasticity (or metaplasticity)
- 2) The rationale to select the agonists/antagonists DCS and memantine are not clear. Why was ifenprodil not used in these experiments (given the GluN2B excess)
- 3) Fluoxetine is not only an antagonist of GluN2B containing NMDARs, its repetitive use also causes its internalization and thus a switch towards synaptic GluN2A-containing receptors (Ampuero et al., 2010), and this is associated with decreased LTP after fluoxetine.
- 4) It also remains unexplained why repetitive fluoxetine treatment (normalizing NMDAR function in KO mice leads to (supposedly) CamKII-dependent hyperphosphorylation in a model in which GluN2B is blocked/decreased (the main source of calcium rise leading to CaMKII activation)
- 5) Maybe a graphical abstract could clarify those aspects that are supported/not supported by experimental results

Reviewer #2 (Remarks to the Author):

The authors examined the role of SAM4 in synaptic transmission, synaptic plasticity and learning and memory using a knockout mouse lacking SAM4. The KO has larger NMDA fEPSPs in the hippocampus suggesting that the presence of SAM4 reduces the NMDA activity. In contrast, the AMPA receptor fEPSPs were similar to wild-type (WT) mice. They examined the KO mice in a battery of behavioral tasks and found that they behaved similar to WT mice except they retained more contextual fear after a one-week delay. Then they used pharmacological treatments to test whether manipulations that altered NMDA receptor levels would reverse the behavioral effects of the SAM4 KO.

While I find the experiments interesting and the data are consistent with their general conclusions, the pharmacological manipulations are inconclusive since the drugs affect more than just NMDA receptors. Therefore, they cannot conclusively say that the behavioral effects are due to the effects on NMDA receptors seen in the SAM2 KO mice.

1. In figure 1A, what is the pretone freezing before giving the CS- in context A? If the difference is only due to differences in contextual fear there should be a large difference in the pretone freezing. How different is the CS+ from the CS-?
2. Figure 2C: How did they measure both the AMPA and NMDA fEPSPs for the correlation shown in 2C? Did you add an antagonist for one receptor and then was it out and add the other receptor blocker? If so, how do you know that the first blocker was completely washed out? Did they make such

measurements repeatedly in the same slices?

3. Was there a difference in the NMDA currents that were not blocked by ifenprodil?

4. Fluoxetine does more than block GluN2B, so how do you know it is the GluN2B effect that is important? Do other GluN2B blockers like ifenprodil have similar effects?

5. It is not clear if they only used males or also included females. If they only used males this should be specified in the abstract and possibly in the title.

6. Why is the effect only seen 48 and not 24 hours later? 24 hr is generally long enough for consolidation.

7. They loosely use the terms consolidation and retention. Why is a response 7 days later considered consolidation and 1-day retention? How are they defining consolidation?

8. Are there findings specific for this synapse? If so, why? Is there a difference in expression of SALM4 at different synapses in the hippocampus?

9. They suggest that SAM4 interacts at the glutamatergic synapses to normally reduce NMDA currents. How do they know that the change in NMDA is not a compensatory change? It is possible that loss of SAM4 has an indirect effect through changes in serotonin transmission, GABAergic inhibition, etc leading to a compensatory increase in NMDAR.

10. For the DCS and memantine experiments I do not agree with their logic. I do not understand why giving an NMDA receptor agonist chronically would be expected to increase NMDA. I would expect down regulation if the receptors being measured are the same ones being stimulated by the agonist. I think the data suggests that the effect of DCS is not from directly affecting the NMDA receptors that they are measuring. Does incubating neuronal cultures with DCS cause an increase in NMDAR over time?

11. The effects of DCS and memantine are variable and difficult to interpret. Also, the behavioral effect did not reach statistical significance. These results are inconclusive at best.

12. Similarly, they suggest that fluoxetine blocks GluN2B leading to a compensatory downregulation of receptors.

13. If in WT mice, fluoxetine improves fear memory consolidation by suppressing NMDAR then why would suppressing NMDAR in KO mice reduce fear memory?

14. I miss the logic of their conclusion starting on line 232.

15. Figure 6 does not repeat the findings of Fig 1. On day 8 the WT and KO vehicle are not different. Instead the difference shows up on days 9 and 10 which probably represent extinction from the context exposure on the previous days. Fluoxetine seems to affect more fear recall and treated animals show little extinction across days.

16. The inability of the double KO to reverse the effects of the SALM4 KO could mean that the effects are not due to inhibiting the other SALMs.

17. Why do their results suggest that SALM4 KO is less likely to involve SALM5?

18. Line 126 is unclear. It says a "ratio of NMDA" but if it is only NMDA it is not a ratio.

Reviewer #3 (Remarks to the Author):

Nature Review November 2020

The experiments described in this manuscript investigate the role of a specific class of synaptic adhesion molecules in modifying synaptic functions and more specifically postsynaptic receptor responses. This set of experiments focused on a specific synaptic adhesion molecule called SALM4/Lrnf3 that, during development inhibits synapse development by suppressing other SALM family proteins.

These investigators were interested in whether SALM4 inhibits synaptic function and learning and memory processes.

The results showed that SALM4-knockout (Lrnf3^{-/-}) mice showed enhanced long-term contextual fear memory consolidation but not acquisition or short-term retention.

Electrophysiological evidence is also presented showing that the hippocampus of the *Lrnf3*^{-/-} mutant mice showed increased currents of GluN2B-containing NMDA receptors but not AMPA receptors.

Chronic treatment of *Lrnf3*^{-/-} mutant mice with fluoxetine, a selective serotonin reuptake inhibitor used to treat mood disorders that amongst other things directly inhibits GluN2B-NMDARs. This effect resulted in normal NMDAR function and long-term contextual fear memory consolidation in the mutant mice.

Taken together, the authors argue that this pattern of results suggest that SALM4 suppresses excessive GluN2B-NMDAR function and long-term fear memory consolidation specifically.

Abstract:

I thought that the abstract was well writing and clear.

Introduction:

-line 67-what do the authors mean by clustering? Are they describing trafficking and insertion processes during plasticity? More detail here is advised.

-how much actual modification of NMDA receptors occurs in comparison to AMPA receptors. I thought the former was more static and the latter very dynamic based on activity patterns elicited by new experiences. The authors need to unpack this to make their position clear.

**key to paper-However, whether SALM4 also negatively regulates synaptic functions such as postsynaptic NMDAR responses and any related behaviors remains unclear.

-line 83-you state "In the present study, we found that SALM4-knockout (*Lrnf3*^{-/-}) mice displayed abnormally increased NMDAR-mediated, but not AMPAR-mediated, synaptic transmission involving the NMDAR GluN2B subunit."

-I didn't think that NMDAR activation resulted in much synaptic transmission (how much contribution do they make to an EPSP for example compared to AMPA).

-line 89-when you say excessive fear memory in humans are you referring to anxiety disorders or PTSD? You should make this clear.

RESULTS and DISCUSSION:

-line 100-enhanced context conditioning 48 hour retention (cue/context version)

-line 102-enhanced 7 day retention (context only version)

*this is a strange difference. How do you account for this? Why enhanced 48 hour retention on both cue and context version?

-your report normal extinction. Shouldn't they be impaired on extinction if the memory is so strong?

-I do not find the context consolidation effect that large.

-line 110-not enough training to see good spatial specificity. This level of performance is probably not even HPC mediated (see Mehla, J., Faraji, J., Mohajerani, M.H., and McDonald, R.J. (2018) Looking beyond the standard version of the Morris water task in the assessment of mouse models of cognitive deficits. Hippocampus).

-I did not see memory consolidation assessments 48 hours and 7 days later to assess memory consolidation on this task.

-line 155-I do not understand why this mechanism specifically enhanced context memory consolidation and not spatial memory? The spatial version of the Morris water task is a "gold standard" for hippocampal function in rodents.

-line 119-the Fanselow citation is used as support for the basically debunked idea of hippocampal/neocortical systems consolidation. The Shimizu citation is interesting but a bit puzzling as rats with complete hippocampal damage can show normal fear conditioning to context (see Lee et al., 2018).

-line 125-the hippocampus is not crucial for fear conditioning to context. That citation and the other Fanselow cite have not been replicated.

-line 145-is it not surprising that LTP is not enhanced or the long-term maintenance (consolidation) is not enhanced in at least the adult mutant mice?

-line 162-Why did memantine manipulations not block LTP in either group?

-line 168- you state "Chronic memantine treatment failed to rescue the enhanced fear memory in Lrnf3-/- mice (Fig. 4c-g)." This sentence should be changed.

-line 172-the explanations of this paradoxical data is not very satisfying.

-line 180-I don't like the logic for this experiment. Sure, Fluoxetine inhibits GluN2B receptors but it has other effects on other neurotransmitter systems as well.

-line 201-acquisition of fear conditioning to context is not dependent on the hippocampus.

-line 220-you state : "Chronic fluoxetine treatment induced strong (> 2-folds) increases in GluA1 Ser-831 and Ser-845 phosphorylation in Lrnf3-/- mice but not in WT mice, without affecting total protein levels.

*So what does this mean? An increased NMDA signal but no effects on LTP and changes in GluA1 (AMPA receptors) serine phosphorylation but no changes in protein levels?

-line 248-the logic and importance of this experiment is not made clear. Another problem with interpreting the double knock-out is you do not assess any thing besides these phosphorylation changes.

Point-by-point responses to reviewers' comments:

Reviewer #1 (Remarks to the Author):

This study addresses the synaptic role of the synaptic cell adhesion-like molecule SALM-4, using the corresponding KO mice. The enhanced GluN2B-mediated NMDAR function associated with enhanced contextual fear memory consolidation indicates that SALM4 participates in the suppression of excessive GluN2B-containing NMDAR function and in fear memory consolidation.

This is an interesting piece of work describing how the SALM-4 KO affects fear memory, glutamate receptor activity and levels without (supposedly) affecting synaptic plasticity, at least LTP at SC to CA1 synapses. It is presented in an organized and clear manner. Moreover, effects of NMDAR agonists and antagonists were tested. In the discussion, some minor issues should be addressed:

→ We appreciate the thoughtful and constructive comments of the reviewer and tried our best to address them.

1) Other forms of synaptic plasticity (LTD) or plasticity in other hippocampal pathways have not been examined. Given that the molecular composition of hippocampal synapses vary in a circuit-dependent manner (and even in a distal vs proximal location, well described at least in the stratum radiatum), it is additionally possible that the SC-CA1 pathway is spared from LTP deficiency while other pathways are affected. Moreover, the increase of GluN2B-mediated synaptic responses should enhance plasticity (or metaplasticity)

→ In response, we performed additional LTP experiments for the TA (temporoammonic)-to-CA1 pathway and found that there is no genotype difference in *Lrfn3*^{-/-} mice (2–3 months) (**Supplementary Fig. 3i–l**), similar to the lack of genotype difference in LTP (HFS-LTP, TBS-LTP) in the SC-to-CA1 pathway (**Fig. 7a–f**). In addition, there was no genotype difference in LTD (SP-LFS-LTD; single-pulse low frequency stimulation-induced LTD) in the SC-to-CA1 pathway (**Fig. 7g,h**); we clarified this in the revised Results. These results suggest that there is no genotype difference in LTD and no pathway-specific differences in the hippocampal synapses that we examined.

2) The rationale to select the agonists/antagonists DCS and memantine are not clear. Why was ifenprodil not used in these experiments (given the GluN2B excess)?

→ We were discouraged from performing experiments using ifenprodil because memantine failed to rescue the fear phenotype in *Lrfn3*^{-/-} mice (**Fig. 4**). However, in response to this comment, we tested whether ifenprodil could rescue the fear phenotype and found that it did not rescue the enhanced fear consolidation in *Lrfn3*^{-/-} mice (**Supplementary Fig. 6**), similar to the effects of memantine. These results

collectively suggest that fluoxetine may rescue the fear phenotype in *Lrfn3*^{-/-} mice through mechanisms, including NMDAR inhibition and serotonin enhancements; this was clarified in the revised Discussion.

3) Fluoxetine is not only an antagonist of GluN2B containing NMDARs, its repetitive use also causes its internalization and thus a switch towards synaptic GluN2A-containing receptors (Ampuero et al., 2010), and this is associated with decreased LTP after fluoxetine.

→ We appreciate this comment and described this aspect with the citation of related references in the revised Results.

4) It also remains unexplained why repetitive fluoxetine treatment (normalizing NMDAR function in KO mice leads to (supposedly) CamKII-dependent hyperphosphorylation in a model in which GluN2B is blocked/decreased (the main source of calcium rise leading to CaMKII activation).

→ We appreciate this comment. It is possible that the changes in phosphatase actions, which are much less understood, might also play a role in the increased GluA1-S831/GluN2B-S1303 phosphorylation. We clarified this in the revised Results with a reference.

5) Maybe a graphical abstract could clarify those aspects that are supported/not supported by experimental results

We appreciate this comment and added a graphical abstract in the last Supplementary Figure (**Supplementary Fig. 10**).

Reviewer #2 (Remarks to the Author):

The authors examined the role of SALM4 in synaptic transmission, synaptic plasticity and learning and memory using a knockout mouse lacking SALM4. The KO has larger NMDA fEPSPs in the hippocampus suggesting that the presence of SALM4 reduces the NMDA activity. In contrast, the AMPA receptor fEPSPs were similar to wild-type (WT) mice. They examined the KO mice in a battery of behavioral tasks and found that they behaved similar to WT mice except they retained more contextual fear after a one-week delay. Then they used pharmacological treatments to test whether manipulations that altered NMDA receptor levels would reverse the behavioral effects of the SAM4 KO. While I find the experiments interesting and the data are consistent with their general conclusions, the pharmacological manipulations are inconclusive since the drugs affect more than just NMDA receptors. Therefore, they cannot conclusively say that the behavioral effects are due to the effects on NMDA receptors seen in the SALM4 KO mice.

→ We appreciate the thoughtful and constructive comments of the reviewer and tried

our best to address them.

1. In figure 1A, what is the pretone freezing before giving the CS- in context A? If the difference is only due to differences in contextual fear there should be a large difference in the pretone freezing. How different is the CS+ from the CS-?

→ The “CS-“ in the context B (24 hours after the fear memory acquisition) and in the context A (48 hours after) indicates “no tone” presentation, meaning the presentation of only a spatial context. As correctly pointed out by the reviewer, in the context A after 48 hours, the freezing levels are high, indicating that the spatial context A properly evoked contextual fear memory recall. We clarified this in the revised Results and figure legend.

2. Figure 2C: How did they measure both the AMPA and NMDA fEPSPs for the correlation shown in 2C? Did you add an antagonist for one receptor and then was it out and add the other receptor blocker? If so, how do you know that the first blocker was completely washed out? Did they make such measurements repeatedly in the same slices?

→ We first measured AMPA fEPSPs in the presence of AP5, and then changed the extracellular/bath solution to that containing NBQX to isolate NMDA fEPSPs. NMDAR-fEPSPs were measured after at least 30-min washout of AP5 and stabilization of NMDAR-fEPSPs. Although we are not sure if AP5 was completely washed out, it is highly likely based on the observed shift and stabilization patterns in fEPSPs. We clarified this in the revised Methods.

3. Was there a difference in the NMDA currents that were not blocked by ifenprodil?

→ There was a genotype difference in the NMDAR currents not blocked by ifenprodil. We now show this data in the revised **Fig. 2b**.

4. Fluoxetine does more than block GluN2B, so how do you know it is the GluN2B effect that is important? Do other GluN2B blockers like ifenprodil have similar effects?

→ In response, we tested whether ifenprodil could rescue the fear phenotype in *Lfn3*^{-/-} mice and found that it did not rescue the enhanced fear consolidation (**Supplementary Fig. 6**), similar to the effects of memantine (**Fig. 4**). Although there could be multiple reasons for this result, we now tone down and comment that the fluoxetine-dependent rescue of the fear phenotype likely involves more than NMDAR inhibition in the revised Results.

5. It is not clear if they only used males or also included females. If they only used males this should be specified in the abstract and possibly in the title.

→ We used only males, and this was clarified in the Abstract.

6. Why is the effect only seen 48 and not 24 hours later? 24 hr is generally long enough

for consolidation.

→ The results at 24 hours actually show a strong tendency, although it did not reach a statistical significance. We decided not to comment on this based on the statistical results.

7. They loosely use the terms consolidation and retention. Why is a response 7 days later considered consolidation and 1-day retention? How are they defining consolidation?

→ This is an important point. Given that a 7-day period is frequently used as a time frame for fear memory consolidation, we changed “48-hr fear memory consolidation” to “48-hr fear memory retention” and added the following sentence to the revised Results: “Here, fear memory lasting for seven days was defined as consolidated memory, based on previous results (Cohen et al., 2006; McGaugh, 2000), although we did not directly test it in the present study by using, i.e., protein synthesis inhibitors.”.

8. Are there findings specific for this synapse? If so, why? Is there a difference in expression of SALM4 at different synapses in the hippocampus?

→ In response, we performed FISH (fluorescent in situ hybridization) experiments and found that SALM4 expression is detected in other hippocampal regions, including the dentate gyrus and CA3 regions (**Supplementary Fig. 4**). We also found that SALM4 is detected in both glutamate and GABA neurons, suggesting the interesting possibility that SALM4 may regulate excitatory synapses in GABA neurons, which will be explored in future studies. This is also in line with the previous results that SALM4 KO leads to increases in the frequency of both excitatory and inhibitory synaptic transmission (Lie et al., 2016).

9. They suggest that SALM4 interacts at the glutamatergic synapses to normally reduce NMDA currents. How do they know that the change in NMDA is not a compensatory change? It is possible that loss of SAM4 has an indirect effect through changes in serotonin transmission, GABAergic inhibition, etc leading to a compensatory increase in NMDAR.

→ In response, we further explored the mechanisms by which SALM4 potentially inhibits NMDAR function. One was, as described in the Discussion section of the original manuscript, that SALM4 interacts in cis with SALM3/5 in the postsynaptic membrane and inhibits SALM3/5's trans-synaptic interaction with presynaptic PTP σ (Choi et al., 2016; Li et al., 2015; Mah et al., 2010), a member of the LAR family receptor protein tyrosine phosphatases, known to promote NMDAR function through as yet unidentified trans-synaptic interaction (Kim et al., 2020; Scip and Sudhof, 2020). If this hypothesis is correct, we should be able to knockdown PTP σ (encoded by *Ptprs*) and block the SALM4 deletion-dependent enhancement of NMDAR function. As expected, we could rescue the increased NMDAR function in the CA1 region of *Lrfn3*^{-/-}

mice by suppressing the expression of PTP σ in CA3 neurons, which is targeted to the presynaptic side of SC-CA1 synapses (**Fig. 9; Supplementary Fig. 9**). We commented on this in Discussion and Abstract in addition to Results.

10. For the DCS and memantine experiments I do not agree with their logic. I do not understand why giving an NMDA receptor agonist chronically would be expected to increase NMDA. I would expect down regulation if the receptors being measured are the same ones being stimulated by the agonist. I think the data suggests that the effect of DCS is not from directly affecting the NMDA receptors that they are measuring. Does incubating neuronal cultures with DCS cause an increase in NMDAR over time?

→ We agree with the reviewer that it is highly likely that NMDAR agonist induces downregulation of NMDAR responses, although the exact responses in WT and mutant mice could differ depending on drug concentrations and treatment schemes. That was one of the reasons why we tried both NMDAR agonist and antagonist in the experiments. We clarified this in the revised Results. We have not attempted DCS treatment in cultured neurons and do not know any previous results to the best of our knowledge.

11. The effects of DCS and memantine are variable and difficult to interpret. Also, the behavioral effect did not reach statistical significance. These results are inconclusive at best.

→ We fully agree with the reviewer and clarified these limitations in the revised Results.

12. Similarly, they suggest that fluoxetine blocks GluN2B leading to a compensatory downregulation of receptors.

→ As pointed out correctly, chronic fluoxetine reduced NMDAR function but also induced unexpected upregulation of signaling pathways downstream of NMDAR activation. It is thus possible that fluoxetine may also act through non-NMDAR mechanisms such as serotonin enhancements; this was clarified in the revised Discussion.

13. If in WT mice, fluoxetine improves fear memory consolidation by suppressing NMDAR then why would suppressing NMDAR in KO mice reduce fear memory?

→ The results from WT mice suggest that 'normal' levels of NMDAR function are required to maintain normal levels of fear memory consolidation. The results from KO mice suggest that 'increased' levels of NMDAR function are involved in abnormally enhancing fear memory consolidation. However, again, these effects could be mediated by non-NMDAR-related mechanisms, as clarified in the revised Discussion.

14. I miss the logic of their conclusion starting on line 232.

→ We modified the sentences to improve the conclusions and also changed the title of this section.

15. Figure 6 does not repeat the findings of Fig 1. On day 8 the WT and KO vehicle are not different. Instead the difference shows up on days 9 and 10 which probably represent extinction from the context exposure on the previous days. Fluoxetine seems to affect more fear recall and treated animals show little extinction across days.

→ We agree that the baseline difference on day 8 in Fig. 6 is not reproduced, as compared with these results in Fig. 1. However, we have to point out that the animals used in Fig. 1 are naïve animals, and those used in Fig. 6 were chronically handled and drug-treated. We clarified this in the revised Results.

Regarding the more significant effect of fluoxetine on fear memory recall on day 8, we agree with the reviewer and added related descriptions to the revised Results.

16. The inability of the double KO to reverse the effects of the SALM4 KO could mean that the effects are not due to inhibiting the other SALMs.

→ It is true for SALM3, but SALM4 also inhibits SALM2 and SALM5. In addition, our new results (mentioned above) indicate that knockdown of presynaptic PTP σ , a presynaptic receptor protein tyrosine phosphatase binding to both postsynaptic SALM3 and SALM5, suppress SALM4-KO-induced NMDAR hyperactivity (**Fig. 9**), suggesting that SALM5 may act through presynaptic PTP σ to regulate NMDAR function. We clarified this in the revised Results and Discussion.

17. Why do their results suggest that SALM4 KO is less likely to involve SALM5?

→ Our sincere apologies for the mistake; “SALM5” was changed to “SALM3”.

18. Line 126 is unclear. It says a “ratio of NMDA” but if it is only NMDA it is not a ratio.

→ Our sincere apologies; “the ratio of NMDAR-mediated” was changed to “NMDAR-mediated”.

Reviewer #3 (Remarks to the Author):

Nature Review November 2020

The experiments described in this manuscript investigate the role of a specific class of synaptic adhesion molecules in modifying synaptic functions and more specifically postsynaptic receptor responses. This set of experiments focused on a specific synaptic adhesion molecule called SALM4/Lrnf3 that, during development inhibits synapse development by suppressing other SALM family proteins.

These investigators were interested in whether SALM4 inhibits synaptic function and learning and memory processes.

The results showed that SALM4-knockout (*Lrfr3^{-/-}*) mice showed enhanced long-term contextual fear memory consolidation but not acquisition or short-term retention.

Electrophysiological evidence is also presented showing that the hippocampus of the *Lrfr3^{-/-}* mutant mice showed increased currents of GluN2B-containing NMDA receptors but not AMPA receptors.

Chronic treatment of *Lrfr3^{-/-}* mutant mice with fluoxetine, a selective serotonin reuptake inhibitor used to treat mood disorders that amongst other things directly inhibits GluN2B-NMDARs. This effect resulted in normal NMDAR function and long-term contextual fear memory consolidation in the mutant mice.

Taken together, the authors argue that this pattern of results suggest that SALM4 suppresses excessive GluN2B-NMDAR function and long-term fear memory consolidation specifically.

→ We appreciate the thoughtful and constructive review comments of the reviewer.

Abstract:

I thought that the abstract was well written and clear.

→ We appreciate it.

Introduction:

-line 67-what do the authors mean by clustering? Are they describing trafficking and insertion processes during plasticity? More detail here is advised.

→ Dendritic clustering of NMDARs and AMPARs by SALM2 means that synaptic localization of these receptors requires SALM2 (Ko et al., 2006). The dendritic clustering of NMDARs by SALM1 indicates the results from the Wang et al. paper that overexpression of SALM1 in cultured neurons increased the clustering of NMDAR on the dendritic surface (Wang et al., 2006). We clarified this in the revised Results.

-how much actual modification of NMDA receptors occurs in comparison to AMPA receptors. I thought the former was more static and the latter very dynamic based on activity patterns elicited by new experiences. The authors need to unpack this to make their position clear.

→ Although it is true that AMPA receptors are more mobile, NMDA receptors do display activity-dependent trafficking in and out of synapses (Lau and Zukin, 2007).

**key to paper-However, whether SALM4 also negatively regulates synaptic functions such as postsynaptic NMDAR responses and any related behaviors remains unclear.

→ During revision, we made a key finding that presynaptic PTP σ (a member of the LAR family receptor protein tyrosine phosphatases that trans-synaptically interacts with postsynaptic SALM3/5, which are known to be suppressed in cis by SALM4) is required for SALM4-dependent inhibition of NMDARs. In brief, knockdown of PTP σ strongly rescued SALM4 deletion-induced NMDAR hyperactivity (see **Fig. 9** for further details).

-line 83-you state “In the present study, we found that SALM4-knockout (*Lrfr3*^{-/-}) mice displayed abnormally increased NMDAR-mediated, but not AMPAR-mediated, synaptic transmission involving the NMDAR GluN2B subunit.”

-I didn't think that NMDAR activation resulted in much synaptic transmission (how much contribution do they make to an EPSP for example compared to AMPA).

→ Although the size of peak currents is relatively small, NMDARs do mediate synaptic currents with slow kinetics, which are significant in size (area under the curve) and have mainly modulatory effects (protein modifications and gene expression) (Hansen et al., 2017).

-line 89-when you say excessive fear memory in humans are you referring to anxiety disorders or PTSD? You should make this clear.

→ We meant PTSD and clarified it in the revised Introduction.

RESULTS and DISCUSSION:

-line 100-enhanced context conditioning 48 hour retention (cue/context version)

-line 102-enhanced 7 day retention (context only version)

*this is a strange difference. How do you account for this? Why enhanced 48 hour retention on both cue and context version?

→ The results actually indicate that only contextual, but not cued, fear memory was enhanced. We clarified this in the revised Results.

-your report normal extinction. Shouldn't they be impaired on extinction if the memory is so strong?

→ It is probably because, on day 1, there is no genotype difference in 24-hr fear memory retention, although there is a tendency for impairment in fear extinction.

-I do not find the context consolidation effect that large.

→ We agree that the effect size is small. However, the results are statistically significant and reproducible. We clarified this in the revised results by modifying

“increased/enhanced” to “moderately increased/enhanced”.

-line 110-not enough training to see good spatial specificity. This level of performance is probably not even HPC mediated (see Mehla, J., Faraji, J., Mohajerani, M.H., and McDonald, R.J. (2018) Looking beyond the standard version of the Morris water task in the assessment of mouse models of cognitive deficits. Hippocampus).

→ We agree with the reviewer and commented on the modest effect and that there is a need to try other versions of the Morris water maze test, suggested by the reviewer, in the revised Results. However, we have to point out that we focused on the lack of clear difference between WT and mutant mice.

-I did not see memory consolidation assessments 48 hours and 7 days later to assess memory consolidation on this task.

→ It is correct. However, given that a 7-day period is frequently used as a time frame for fear memory consolidation, we changed “48-hr fear memory consolidation” to “48-hr fear memory retention” and added the following sentence to the revised Results: “Here, fear memory lasting for seven days was defined as consolidated memory, based on previous results (Cohen *et al.*, 2006; McGaugh, 2000), although we did not directly test it in the present study by using, i.e., protein synthesis inhibitors.”.

-line 155-I do not understand why this mechanism specifically enhanced context memory consolidation and not spatial memory? The spatial version of the Morris water task is a “gold standard” for hippocampal function in rodents.

→ We agree. However, the two behavioral tests are not identical and likely involve different neural mechanisms. We have to point out that it is not uncommon that different memory impairments are observed in the same mouse line.

-line 119-the Fanselow citation is used as support for the basically debunked idea of hippocampal/neocortical systems consolidation. The Shimizu citation is interesting but a bit puzzling as rats with complete hippocampal damage can show normal fear conditioning to context (see Lee *et al.*, 2018).

→ We appreciate this comment and removed these two references that are not fully relevant to the conclusions from this section of the manuscript.

-line 125-the hippocampus is not crucial for fear conditioning to context. That citation and the other Fanselow cite have not been replicated.

→ We corrected the text as follows: “...hippocampus, a brain region critical for contextual fear memory” was changed to “...hippocampus, a brain region that, together with the amygdala, contributes to contextual fear memory”. In addition, we replaced the Kim *et al* paper with more recent ones pointing to the importance of the coordinated activity of the amygdala and hippocampus (Kim and Jung, 2006; Maren *et al.*, 2013;

Tovote et al., 2015).

-line 145-is it not surprising that LTP is not enhanced or the long-term maintenance (consolidation) is not enhanced in at least the adult mutant mice?

→ We agree. However, it is possible that some compensatory changes might have happened. In support of this possibility, our results indicate that chronic fluoxetine treatment in adult mice strongly suppresses LTP in WT mice but not in SALM4-KO mice, likely through abnormally upregulated LTP-related downstream mechanisms (i.e., increased GluA1 phosphorylation) (**Figs. 7 and 8**). This was clarified in the revised Results.

-line 162-Why did memantine manipulations not block LTP in either group?

→ This is an important question. The difference between memantine and D-cycloserine on their effects on NMDAR currents could be attributable to that they act through different mechanisms; memantine is an open-channel blocker, and D-cycloserine is a glycine-site antagonist (Chen et al., 1992). Alternatively, it could be because they act on different neuronal types; i.e. memantine acts more strongly on NMDARs on GABA neurons (Povysheva and Johnson, 2016). We clarified this in the revised Results as follows: “The distinct effects of DCS and memantine might involve differences in dosage/treatment scheme, action mechanisms (glycine-site antagonist and open-channel blocker, respectively), or sensitivity of distinct neuronal types and synapses (Chen and Lipton, 1997; Lipton, 2006; Povysheva and Johnson, 2016).”.

-line 168- you state “Chronic memantine treatment failed to rescue the enhanced fear memory in *Lrn3*^{-/-} mice (Fig. 4c–g).” This sentence should be changed.

→ We changed “failed to rescue” to “did not rescue”.

-line 172-the explanations of this paradoxical data is not very satisfying.

→ We changed the conclusions of the section to be more statistically correct and as accurately as possible.

-line 180-I don't like the logic for this experiment. Sure, fluoxetine inhibits GluN2B receptors but it has other effects on other neurotransmitter systems as well.

→ We agree and corrected the sentence to comment on that fluoxetine could also act as a selective serotonin reuptake inhibitor in addition to NMDAR antagonist.

-line 201-acquisition of fear conditioning to context is not dependent on the hippocampus.

→ We changed the text in the revised Results to highlight that LTP processes in the hippocampus and amygdala are associated with fear memory acquisition/consolidation.

-line 220-you state : “Chronic fluoxetine treatment induced strong (> 2-folds) increases in GluA1 Ser-831 and Ser-845 phosphorylation in *Lrnf3*^{-/-} mice but not in WT mice, without affecting total protein levels.

*So what does this mean? An increased NMDA signal but no effects on LTP and changes in GluA1 (AMPA receptors) serine phosphorylation but no changes in protein levels?

→ We hypothesize that the decreased NMDAR currents in the KO mice chronically treated with fluoxetine lead to compensatory mechanisms such as LTP-promoting mechanisms to prevent the decrease in LTP from happening while inducing some changes downstream of NMDAR activation such as posttranslational modification of synaptic/neuronal proteins or changes in gene expression, as described in Discussion.

-line 248-the logic and importance of this experiment is not made clear. Another problem with interpreting the double knock-out is you do not assess anything besides these phosphorylation changes.

→ The unclear logic was because we wrote down SALM5 instead of SALM3. We apologize for the mistake and have corrected it to SALM3. In addition, because the double knock-out did not rescue the enhanced fear memory consolidation, we did not attempt additional experiments such as biochemical and synaptic characterizations. We clarified this in the revised Results.

References

- Chen, H.S., and Lipton, S.A. (1997). Mechanism of memantine block of NMDA-activated channels in rat retinal ganglion cells: uncompetitive antagonism. *J Physiol* 499 (Pt 1), 27-46.
- Chen, H.S., Pellegrini, J.W., Aggarwal, S.K., Lei, S.Z., Warach, S., Jensen, F.E., and Lipton, S.A. (1992). Open-channel block of N-methyl-D-aspartate (NMDA) responses by memantine: therapeutic advantage against NMDA receptor-mediated neurotoxicity. *J Neurosci* 12, 4427-4436.
- Choi, Y., Nam, J., Whitcomb, D.J., Song, Y.S., Kim, D., Jeon, S., Um, J.W., Lee, S.G., Woo, J., Kwon, S.K., et al. (2016). SALM5 trans-synaptically interacts with LAR-RPTPs in a splicing-dependent manner to regulate synapse development. *Sci Rep* 6, 26676. 10.1038/srep26676.
- Cohen, H., Kaplan, Z., Matar, M.A., Loewenthal, U., Kozlovsky, N., and Zohar, J. (2006). Anisomycin, a protein synthesis inhibitor, disrupts traumatic memory consolidation and attenuates posttraumatic stress response in rats. *Biol Psychiatry* 60, 767-776. 10.1016/j.biopsych.2006.03.013.
- Hansen, K.B., Yi, F., Perszyk, R.E., Menniti, F.S., and Traynelis, S.F. (2017). NMDA Receptors in the Central Nervous System. *Methods Mol Biol* 1677, 1-80. 10.1007/978-1-4939-7321-7_1.
- Kim, J.J., and Jung, M.W. (2006). Neural circuits and mechanisms involved in Pavlovian fear conditioning: a critical review. *Neurosci Biobehav Rev* 30, 188-202.

- 10.1016/j.neubiorev.2005.06.005.
- Kim, K., Shin, W., Kang, M., Lee, S., Kim, D., Kang, R., Jung, Y., Cho, Y., Yang, E., Kim, H., et al. (2020). Presynaptic PTPsigma regulates postsynaptic NMDA receptor function through direct adhesion-independent mechanisms. *Elife* 9, 10.7554/eLife.54224.
- Ko, J., Kim, S., Chung, H.S., Kim, K., Han, K., Kim, H., Jun, H., Kaang, B.K., and Kim, E. (2006). SALM synaptic cell adhesion-like molecules regulate the differentiation of excitatory synapses. *Neuron* 50, 233-245.
- Lau, C.G., and Zukin, R.S. (2007). NMDA receptor trafficking in synaptic plasticity and neuropsychiatric disorders. *Nat Rev Neurosci* 8, 413-426. nrn2153 [pii] 10.1038/nrn2153.
- Li, Y., Zhang, P., Choi, T.Y., Park, S.K., Park, H., Lee, E.J., Lee, D., Roh, J.D., Mah, W., Kim, R., et al. (2015). Splicing-Dependent Trans-synaptic SALM3-LAR-RPTP Interactions Regulate Excitatory Synapse Development and Locomotion. *Cell Rep* 12, 1618-1630. 10.1016/j.celrep.2015.08.002.
- Lie, E., Ko, J.S., Choi, S.Y., Roh, J.D., Cho, Y.S., Noh, R., Kim, D., Li, Y., Kang, H., Choi, T.Y., et al. (2016). SALM4 suppresses excitatory synapse development by cis-inhibiting trans-synaptic SALM3-LAR adhesion. *Nat Commun* 7, 12328. 10.1038/ncomms12328.
- Lipton, S.A. (2006). Paradigm shift in neuroprotection by NMDA receptor blockade: memantine and beyond. *Nat Rev Drug Discov* 5, 160-170. 10.1038/nrd1958.
- Mah, W., Ko, J., Nam, J., Han, K., Chung, W.S., and Kim, E. (2010). Selected SALM (synaptic adhesion-like molecule) family proteins regulate synapse formation. *J Neurosci* 30, 5559-5568. 30/16/5559 [pii] 10.1523/JNEUROSCI.4839-09.2010.
- Maren, S., Phan, K.L., and Liberzon, I. (2013). The contextual brain: implications for fear conditioning, extinction and psychopathology. *Nature Reviews Neuroscience* 14, 417-428. 10.1038/nrn3492.
- McGaugh, J.L. (2000). Memory--a century of consolidation. *Science* 287, 248-251. 10.1126/science.287.5451.248.
- Povysheva, N.V., and Johnson, J.W. (2016). Effects of memantine on the excitation-inhibition balance in prefrontal cortex. *Neurobiol Dis* 96, 75-83. 10.1016/j.nbd.2016.08.006.
- Sclip, A., and Sudhof, T.C. (2020). LAR receptor phospho-tyrosine phosphatases regulate NMDA-receptor responses. *Elife* 9. 10.7554/eLife.53406.
- Tovote, P., Fadok, J.P., and Luthi, A. (2015). Neuronal circuits for fear and anxiety. *Nat Rev Neurosci* 16, 317-331. 10.1038/nrn3945.
- Wang, C.Y., Chang, K., Petralia, R.S., Wang, Y.X., Seabold, G.K., and Wenthold, R.J. (2006). A novel family of adhesion-like molecules that interacts with the NMDA receptor. *The Journal of neuroscience : the official journal of the Society for Neuroscience* 26, 2174-2183.

REVIEWERS' COMMENTS:

Reviewer #1 (Remarks to the Author):

In the present paper, it was found that SALM4-knockout (*Lrnf3*^{-/-}) mice displayed increased GluN2B containing NMDAR-mediated synaptic responses involving additionally the presynaptic receptor tyrosine phosphatase PTP σ , and this is associated with a specific cognitive effect, i.e. disturbed contextual fear memory consolidation, but not with other cognitive defects. The paper has improved considerably during the review process and all the comments raised by this reviewer have been carefully addressed. The results are of high interest considering the following general points:
A specific role for one family member of the synaptic adhesion molecules SALMs is revealed
A specific effect on memory is caused in the KO, highlighting that the SALM genotype, including possible mutations, may affect highly specific behaviors
The behavioral effect and synaptic over-activity can be rescued by a clinically relevant antidepressant therapeutic agent. It remains to be determined in the future whether other SSRIs (with different pharmacological profiles regarding NMDARs) have the same effect.
The paper represents an advance in understanding the molecular mechanism involved in SALM4 function, e.g. involving the presynaptic PTP σ

Thus, I recommend to publish the manuscript in its actual version

Reviewer #2 (Remarks to the Author):

In general, the authors were very responsive to the critiques and the manuscript is more complete. However, there are still a few issues that could be further improved.

1. The abstract still seems to push that the effect of fluoxetine is through GluN2B direct block. The authors use ifenprodil to show changes in GluN2B and yet ifenprodil does not have the same effect as fluoxetine. This seems to suggest that at least GluN2B blockade alone is insufficient to produce the behavioral effect. At worst, this could mean that the GluN2B block is not even involved in the behavioral change.
2. I am still not sure if it is clear that all experiments were done only on male mice. In the revision, I realized that there is no animal section that clearly states the types of animals used in the experiments and how they were obtained and housed even in the methods section. This section should be added and they should simply indicate that all experiments were done with male mice.
3. Was the freezing on day 8 in figure 4D different between the wt and KO vehicle groups?
4. Since the non-GluN2B NMDAR currents also decreased, how do they know that the behavioral difference is not from the decreased non-GluN2B rather than the increased GluN2B? Did fluoxetine not affect the NMDAR currents that were not blocked by ifenprodil? Since Ampuero et al., 2010 found that chronic fluoxetine increases GluN2A, then the change in GluN2A rather than GluN2B could be more important for the behavioral change seen after chronic fluoxetine.
5. Starting on line 171 in the results they still say that the expected results were seen with an increase in NMDA after giving the chronic agonist and a decrease after giving the antagonist. If they agree that giving a NMDAR agonist is likely to downregulate NMDAR responses then why would they expect an increase in NMDAR responses? I also did not see any discussion about why they might have seen an increase in NMDAR responses after giving a chronic NMDAR agonist and a decrease after giving a chronic antagonist. They make it sound like the result is obvious, but the logic is not obvious.

Reviewer #3 (Remarks to the Author):

I was one of the original reviewers and I am happy with the responses to my concerns.

Aug 21, 2021

Re: **COMMSBIO-20-2989A**

SALM4 negatively regulates NMDA receptor function and fear memory consolidation

Author responses to reviewers' comments

REVIEWERS' COMMENTS:

Reviewer #1 (Remarks to the Author):

In the present paper, it was found that SALM4-knockout (*Lrnf3*^{-/-}) mice displayed increased GluN2B containing NMDAR-mediated synaptic responses involving additionally the presynaptic receptor tyrosine phosphatase PTP σ , and this is associated with a specific cognitive effect, i.e. disturbed contextual fear memory consolidation, but not with other cognitive defects. The paper has improved considerably during the review process and all the comments raised by this reviewer have been carefully addressed.

The results are of high interest considering the following general points:

A specific role for one family member of the synaptic adhesion molecules SALMs is revealed

A specific effect on memory is caused in the KO, highlighting that the SALM genotype, including possible mutations, may affect highly specific behaviors

The behavioral effect and synaptic over-activity can be rescued by a clinically relevant antidepressant therapeutic agent. It remains to be determined in the future whether other SSRIs (with different pharmacological profiles regarding NMDARs) have the same effect.

The paper represents an advance in understanding the molecular mechanism involved in SALM4 function, e.g. involving the presynaptic PTP σ

Thus, I recommend to publish the manuscript in its actual version

→ We appreciate the final comments of the reviewer.

Reviewer #2 (Remarks to the Author):

In general, the authors were very responsive to the critiques and the manuscript is more complete. However, there are still a few issues that could be further improved.

1. The abstract still seems to push that the effect of fluoxetine is through GluN2B direct block. The authors use ifenprodil to show changes in GluN2B and yet ifenprodil does not have the same effect as fluoxetine. This seems to suggest that at least GluN2B blockade alone is insufficient to produce the behavioral effect. At worst, this could mean

that the GluN2B block is not even involved in the behavioral change.

→ We agree with the reviewer and tone down the claim in Abstract that GluN2B is the major mediator of behavioral impairment in SALM4 KO mice as follows: “Chronic treatment of *Lrfn3*^{-/-} mice with fluoxetine, a selective serotonin reuptake inhibitor used to treat excessive fear memory that directly inhibits GluN2B-NMDARs, normalizes NMDAR function and contextual fear memory consolidation in *Lrfn3*^{-/-} mice” was changed to “Chronic treatment of *Lrfn3*^{-/-} mice with fluoxetine, a selective serotonin reuptake inhibitor used to treat excessive fear memory that directly inhibits GluN2B-NMDARs, normalizes NMDAR function and contextual fear memory consolidation in *Lrfn3*^{-/-} mice, although the GluN2B-specific NMDAR antagonist ifenprodil was not sufficient to reverse the enhanced fear memory consolidation.”.

2. I am still not sure if it is clear that all experiments were done only on male mice. In the revision, I realized that there is no animal section that clearly states the types of animals used in the experiments and how they were obtained and housed even in the methods section. This section should be added and they should simply indicate that all experiments were done with male mice.

→ Our apologies for the details. We added an “Animals” section in the revised manuscript where we now describe the details on the SALM4/*Lrfn3*-KO mice, animal maintenance conditions, institutional committee/approval, and the sex of animals (all males).

3. Was the freezing on day 8 in figure 4D different between the wt and KO vehicle groups?

→ Unfortunately, we cannot obtain the answer statistically because the results of the two-way ANOVA analysis indicate that only genotype p value is significant whereas interaction p value is not, thus not allowing a posthoc comparison. However, if we perform a Student's t-test, there is no significant genotype difference on day 8 likely because of mouse handling/injection effects.

4. Since the non-GluN2B NMDAR currents also decreased, how do they know that the behavioral difference is not from the decreased non-GluN2B rather than the increased GluN2B? Did fluoxetine not affect the NMDAR currents that were not blocked by ifenprodil? Since Ampuero et al., 2010 found that chronic fluoxetine increases GluN2A, then the change in GluN2A rather than GluN2B could be more important for the behavioral change seen after chronic fluoxetine.

→ A quantitative analysis of the changes in NMDAR currents indicates that the ~53% increase in total NMDAR currents can be explained mostly (by the factor 5:1) by the increase of GluN2B-NMDAR currents (150% increase) rather than by the decrease in GluN2A-NMDAR currents (24% decrease), based on the following calculation; $100 \times 33\% = 33$ vs. $153 \times 53\% = 81$ for GluN2B-NMDAR currents [150% increase]; $100 \times 67\% = 67$ vs. $153 \times 33\% = 51$ [24% decrease]]. We clarified this in Results by changing the

relevant sentence as follows: "...an increase in currents mediated by GluN2B-containing NMDARs (~50%), as shown by the increased sensitivity of the mutant currents to the GluN2B-specific inhibitor, ifenprodil (ifenprodil-sensitive NMDAR currents being 33% vs. 53% of total currents in wild-type (WT) and mutant neurons, respectively) (Fig. 2b)." was changed to "...an increase in currents mediated by GluN2B-containing NMDARs (~50%), as shown by the increased sensitivity of the mutant currents to the GluN2B-specific inhibitor, ifenprodil (ifenprodil-sensitive NMDAR currents being 33% vs. 53% of total currents in wild-type (WT) and mutant neurons, respectively) (Fig. 2b), which indicates a 150% increase in GluN2B-NMDAR currents but a 25% decrease in GluN2A-NMDAR currents.".

In addition, fluoxetine is selective for GluN2B-NMDAR relative to GluN2A-NMDAR currents, which has already been clarified in Results as follows; "In addition, fluoxetine directly inhibits GluN2B-containing, but not GluN2A-containing, NMDARs (Kiss et al., 2012; Szasz et al., 2007), although it also acts as a selective serotonin reuptake inhibitor." To further clarify these aspects, we added the following sentence to the second paragraph of Discussion: "A quantitative analysis indicates that the increase in GluN2B-NMDAR currents is greater than the decrease in GluN2A-NMDAR currents by a factor of five, suggesting that the increase in GluN2B-NMDAR currents plays major roles in enhancing contextual fear memory consolidation, although we cannot exclude the possibility that the decreased GluN2A-NMDAR currents may also play a role."

5. Starting on line 171 in the results they still say that the expected results were seen with an increase in NMDA after giving the chronic agonist and a decrease after giving the antagonist. If they agree that giving a NMDAR agonist is likely to downregulate NMDAR responses then why would they expect an increase in NMDAR responses? I also did not see any discussion about why they might have seen an increase in NMDAR responses after giving a chronic NMDAR agonist and a decrease after giving a chronic antagonist. They make it sound like the result is obvious, but the logic is not obvious.

→ To clarify the logic, we changed two relevant sentences in Results as follows: "...In addition, we used both DCS and memantine because chronic drug treatments could lead to opposite responses through receptor desensitization or compensatory responses. At WT synapses, chronic DCS and memantine treatments induced the expected trends (an increase and a decrease, respectively) in NMDAR currents," was changed to "In addition, we used both DCS and memantine because there was the possibility that chronic drug treatments could lead to opposite responses through receptor desensitization or compensatory responses. At WT synapses, chronic DCS and memantine treatments induced ~~the expected~~ trends (an increase and a decrease, respectively) in NMDAR currents."

Reviewer #3 (Remarks to the Author):

I was one of the original reviewers and I am happy with the responses to my concerns.

→ We appreciate the final comments of the reviewer.

References:

- Kiss, J.P., Szasz, B.K., Fodor, L., Mike, A., Lenkey, N., Kurko, D., Nagy, J., and Vizi, E.S. (2012). GluN2B-containing NMDA receptors as possible targets for the neuroprotective and antidepressant effects of fluoxetine. *Neurochem Int* 60, 170-176. 10.1016/j.neuint.2011.12.005.
- Szasz, B.K., Mike, A., Karoly, R., Gerevich, Z., Illes, P., Vizi, E.S., and Kiss, J.P. (2007). Direct inhibitory effect of fluoxetine on N-methyl-D-aspartate receptors in the central nervous system. *Biol Psychiatry* 62, 1303-1309. 10.1016/j.biopsych.2007.04.014.